# Reliable bi-functional nickel-phosphate /TiO$_2$ integration enables stable n-GaAs photo-anode for water oxidation under alkaline condition

Maheswari Arunachalam [1], Rohini Subhash Kanase [2], Kai Zhu [3] ✉ & Soon Hyung Kang [1] ✉

Hydrogen is one of the most widely used essential chemicals worldwide, and it is also employed in the production of many other chemicals, especially carbon-free energy fuels produced via photoelectrochemical (PEC) water splitting. At present, gallium arsenide represents the most efficient photo-anode material for PEC water oxidation, but it is known to either be anodically photocorroded or photopassivated by native metal oxides in the competitive reaction, limiting efficiency and stability. Here, we report chemically etched GaAs that is decorated with thin titanium dioxide (~30 nm-thick, crystalline) surface passivation layer along with nickel-phosphate (Ni-Pi) cocatalyst as a surface hole-sink layer. The integration of Ni-Pi bifunctional co-catalyst results in a highly efficient GaAs electrode with a ~ 100 mV cathodic shift of the onset potential. In this work, the electrode also has enhanced photostability under 110 h testing for PEC water oxidation at a steady current density $J_{ph} > 25$ mA·cm$^{-2}$. The Et-GaAs/TiO$_2$/Ni-Pi $\|$ Ni-Pi tandem configuration results in the best unassisted bias-free water splitting device with the highest $J_{ph}$ (~7.6 mA·cm$^{-2}$) and a stable solar-to-hydrogen conversion efficiency of 9.5%.

Hydrogen (H$_2$) production is a highly energy-intensive process that consumes 95% of the world's fossil-derived electrical energy and about 5% of the world's renewable energy[1,2]. Around 10 million metric tons of hydrogen are produced per year, reflecting the enormous demand for this chemical in a variety of applications, such as fuels and many other industrial processes[3]. In addition to the well-known production methods, the photoelectrochemical (PEC) technique is of considerable interest because it directly converts solar energy into H$_2$ from water[4,5]. The PEC water splitting reactions—including water oxidation and reduction—depend largely on the structure, components, and surface morphology of the photoelectrode. Solar-to-hydrogen conversion efficiencies (STH, %) have been far from industrial requirements for

decades, although remarkable advances have been made in understanding photoactive materials and enhancing their ability to perform the water-splitting reaction[6,7]. Due to an inadequate band edge position and bandgap for PEC water oxidation, many semiconductors suffer from poor solar light absorption, inefficient charge separation/transfer, and unavoidable e$^-$/h$^+$ recombination at the heterojunction interface; thus, there is a need to find ideal photoelectrode candidates with minimal limitations under PEC working conditions[8,9]. Although many photoanode materials (e.g., silicon [Si], gallium arsenide [GaAs], and gallium phosphide [GaP]) have valence-band edges at more negative potentials than metal oxides, as well as typically having optimal bandgaps for efficient solar-driven water splitting, these

[1]Department of Chemistry Education and Optoelectronic Convergence Research Center, Chonnam National University, Gwangju 61186, Republic of Korea. [2]Department of Interdisciplinary Program for Photonic Engineering, Chonnam National University, Gwangju 61186, Republic of Korea. [3]Chemistry and Nanoscience Center, National Renewable Energy Laboratory, Golden, CO 80401, USA. ✉e-mail: Kai.Zhu@nrel.gov; skang@jnu.ac.kr

semiconductors are generally unstable when operated under photo-anodic conditions in aqueous electrolyte[10–12]. Specifically, when oxidizing water to $O_2$, these materials are either anodically photo-corroded or photopassivated by native metal oxides in the competitive reaction[5,13].

In particular, GaAs ($E_g$: 1.4 eV) has a remarkable $e^-$/$h^+$ mobility and requires small positive onset potentials to drive water oxidation, owing to its excellent optical properties and suitable band potentials[14,15]. However, the rapid surface charge recombination lowers the overall STH efficiency[16]. The GaAs surface is partially photoelectrochemically dissolved and etched by photo-generating holes rather than oxidation water. To overcome this drawback, the protective or blocking layers can be used to avoid direct contact between the electrolyte and the sensible photoabsorber materials[17]. There are several requirements for an ideal protective layer, including that the fabrication routes should not damage or significantly modify the intrinsic material properties. In addition, the protective layer needs to be conductive, transparent, and chemically stable along with good mechanical adhesion. To enable efficient and stable photoanode operation, a simple, easy process must be available for making thin, conformal protective layers[18].

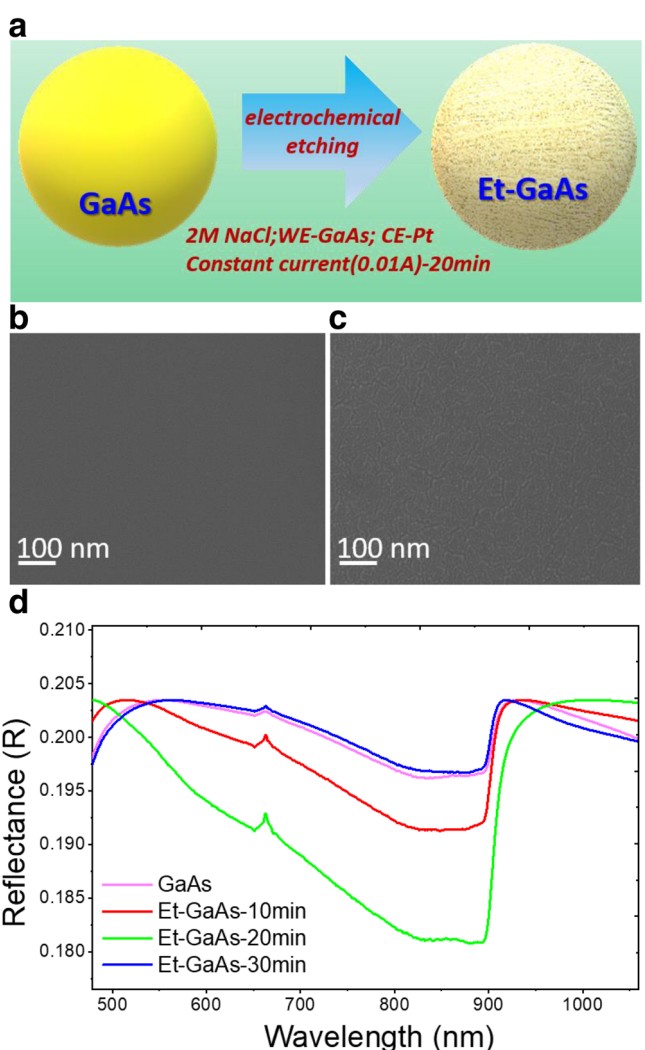

**Fig. 1 | Morphological and optical characterizations of the GaAs and Et-GaAs films. a** Schematic representation of the electrochemical etching condition and the GaAs film after etching (Et-GaAs) directly on planar n-type GaAs wafer. **b**, **c** Surface scanning electron microscopy views of bare GaAs and Et-GaAs film. **d** UV–Vis diffusive reflectance spectra of GaAs and Et-GaAs films for etching times of 10, 20, and 30 min.

Recently, a GaAs photoanode was stabilized using surface pro-tection oxides with the precious metal electrocatalysts, eventually promoting the reaction kinetics[19–21]. Conducting polymer films such as polypyrrole, polystyrene, and polythiophene have been coated on GaAs, but the stability of these systems is limited by peeling of the film, and the electrodes reveal active photocorrosion[22,23]. Shao-Horn et al. reported an n/p-GaAs (001) photocathode that operated in neural pH, stabilized by an epitaxial $SrTiO_3$ surface layer, to deliver a photo-current of 3.1 mA·cm$^{-2}$ at 0.18 V with 24 h stability[24].

In this work, we have achieved solar-light-driven water splitting using chemically etched GaAs decorated with thin $TiO_2$ as the surface passivation layer and a Ni-Pi cocatalyst as the surface hole-sink layer. Chemical etching of GaAs forms high density-graded surface nano-texturing, which reduces light backscattering in the planar water–GaAs interface and maximizes the photo-absorption from the illuminated light. Further, we show that the use of a simple, spin-coated, ~30 nm-thick polycrystalline $TiO_2$ layer can protect the surface of the GaAs photocathode and provide photostability for > 50 h in 1.0 M NaOH under one-sun illumination. Finally, the optimized cocatalyst/protec-tive layer/semiconductor architecture presents a $J_{ph}$ of ~25.77 mA·cm$^{-2}$ and durability of > 100 h in 1.0 M NaOH.

## Results and Discussion
### Basic properties of bare GaAs and etched GaAs (Et-GaAs) pho-toanode films: morphology, crystallinity, optical properties, and chemical information
Figure 1a shows a simple drawing of the electrochemical etching pro-cess, from bare to porous etched GaAs (Et-GaAs), under the mild condition. The Et-GaAs is covered by a layer of nanoporous structure, which has several attractive properties, including low reflectance, an active surface area, and surface-feature sizes at a nanometer scale[25]. Fig. 1b, c shows the surface scanning electron microscopy (SEM) ima-ges of GaAs and Et-GaAs-20 min films. In addition, Supplementary Fig. 1 compared the surface morphology of GaAs films as a function of etching time from 10 min to 30 min and the corresponding etching depth is determined from ~15 to 60 nm by alfa-step profiler as shown in Supplementary Fig. 2. Based on the high $J_{ph}$ and reproducible results in PEC condition, we take the Et-GaAs-20min photoanode for the further surface engineering.

As shown in Fig. 1, the bare GaAs film has a plain and smooth compact surface (Fig. 1b), and the Et-GaAs film has a nanoporous cracked surface morphology (Fig. 1c). We optimized the etching time on the basis of maximizing light absorption with minimal charge recombination loss to give a proper nanotexturing. We evaluated the light absorption properties of Et-GaAs films for different etching times using UV–Vis diffusive reflectance spectra (Fig. 1d). A sharp edge at 924 nm, corresponding to the bandgap of GaAs ($E_g$ = 1.4 eV), was observed. These are quite close to that of bare GaAs, whereas the significant curve changes observed in the wavelength range of 500–900 nm are caused by the etching effect. This may be coming from the light scattering effect, causing it to travel longer paths within the GaAs film inside as well as the increased surface area for the light absorption. This increased path length enhances the probability of photon absorption, enabling to improve the overall light absorption. However, the rapid decrease of light absorption after 30 min etching can be ascribed to the thick native oxides on the surface, composed of an amorphous layer that shields the original absorption via the entire wavelengths. Furthermore, the formation of native oxides ($AsO_x$ and $GaO_x$) is further verified by X-ray diffraction (XRD) and X-ray photo-electron spectroscopy (XPS) results (Supplementary Fig. 3a, b–d).

### Photoelectrochemical properties of bare GaAs and etched GaAs photoanode films
Figure 2a shows the photocurrent density *(J)*–potential *(V)* curves obtained via linear sweep voltammetry from −0.45 to 1.3 $V_{RHE}$ for GaAs

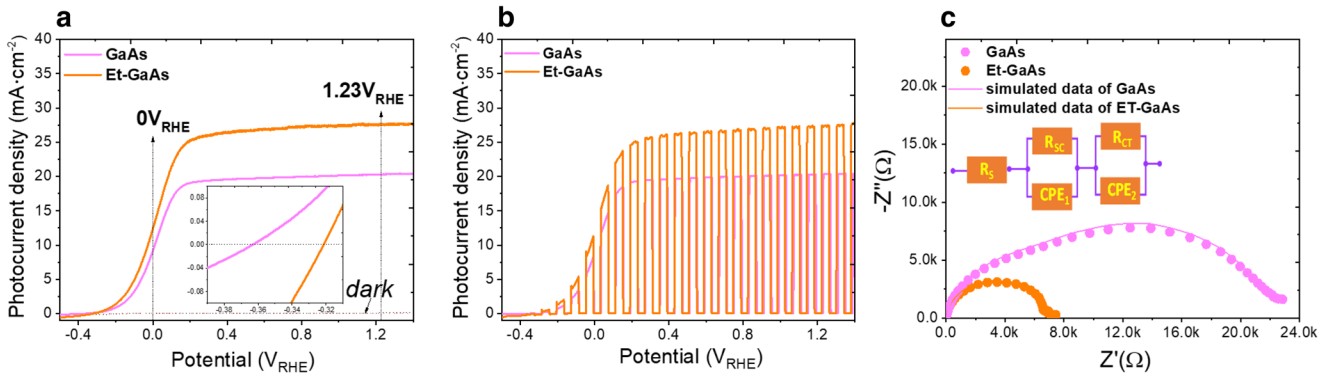

**Fig. 2 | Photoelectrochemical performance and charge transfer kinetics. a** Linear sweep voltammetry curves and (**b**) chopped LSV curves of GaAs and Et-GaAs films. **c** EIS Nyquist plots under illumination at open-circuit voltage condition.

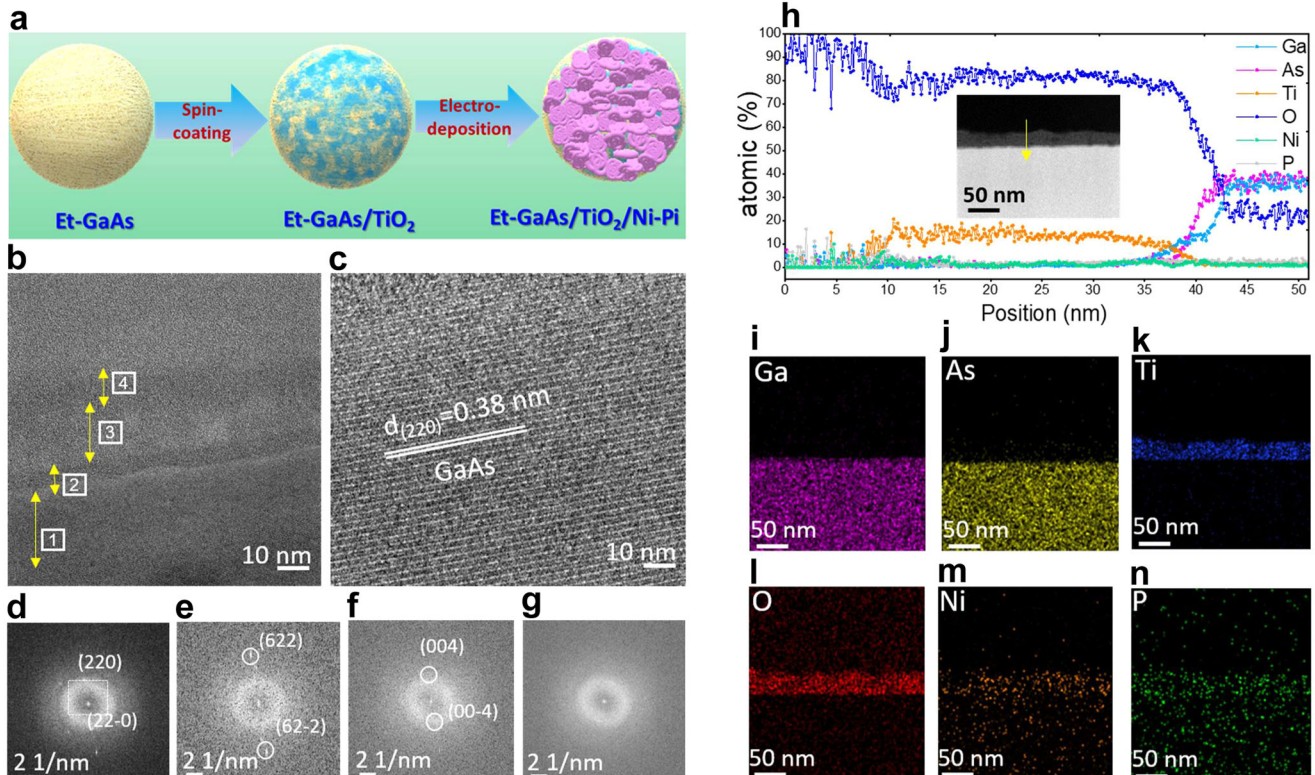

**Fig. 3 | Synthesis and structure of the Et-GaAs/TiO₂/Ni-Pi photoanode.**
**a** Schematic representation of TiO₂ passivation of Et-GaAs film and Ni-Pi deposition of ET-GaAs/TiO₂ film. **b**, **c** Cross-sectional HR-TEM images of Et-GaAs/TiO₂/Ni-Pi

film. **d**–**g** Selected area electron diffraction pattern images of marked areas 1–4 in Fig. 3b. **h** In-depth EDX line scanning of the elements. **i**–**n** Elemental mapping images of Ga, As, Ti, O, Ni, and P, respectively.

and Et-GaAs photoanode films. The Et-GaAs exhibited a positive shift in onset potential, indicating that the surface kinetic barrier was predominant, due to the formation of native oxide during the etching process. The Et-GaAs also showed a significant increase in photocurrent density. The bare GaAs exhibited an onset of photocurrent at a potential of approximately −0.36 $V_{RHE}$, and the best photocurrent densities achieved for GaAs and Et-GaAs were -8.95, 11.28 mA·cm⁻² and 20.34, 27.56 mA·cm⁻² at 0 V and 1.23 $V_{RHE}$, respectively. The effect of the etching time on PEC performance with onset potential was studied and shown in Supplementary Fig. 4. These results demonstrate that the optimum surface etching of the photoanode film is a viable approach to enhance the PEC activity of GaAs-based photoelectrodes. Figure 2b shows the LSV curves of the GaAs and Et-GaAs films under chopped on/

off light illumination at a constant time interval. All samples display the kinetically rapid photo response at all potentials, exhibiting no steady increase of dark current and no degraded shape of the rectangular photocurrent curve. The stability and decay rates of the bare GaAs films measured at different applied potentials are shown in Supplementary Fig. 5.

### Comparison of bare GaAs, Et-GaAs, Et-GaAs/TiO₂, and Et-GaAs/TiO₂/Ni-Pi films: morphological, chemical, and photoelectrochemical properties
Figure 3a illustrates the sequential deposition of TiO₂ and Ni-Pi as a cocatalysts on the Et-GaAs film. We investigated the microstructure, interfacial thickness, and spatial element distribution of the Et-GaAs/

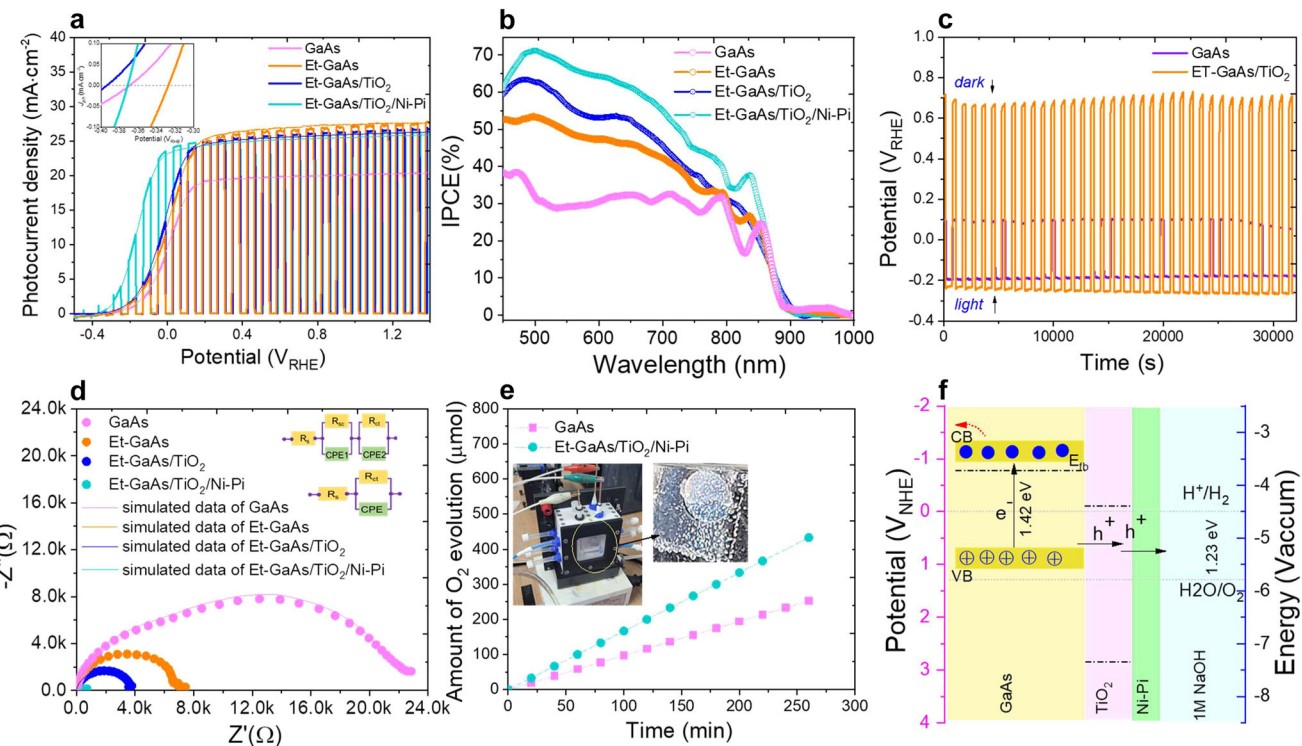

**Fig. 4 | Photoelectrochemical performance of the photoanode film. a** LSV and chopped LSV. **b** Incident photon-to-current efficiency spectra (at 0 $V_{RHE}$). **c** Open-circuit voltage plot in light and dark over a long period of time. **d** EIS Nyquist plots under illumination at open-circuit voltage condition. **e** $O_2$ evolution rate, demonstrated by gas chromatography of GaAs, Et-GaAs, Et-GaAs/TiO$_2$, and Et-GaAs/TiO$_2$/Ni-Pi photoanode films. **f** Representative scheme based on the band alignment depicting the charge transfer/transport of the Et-GaAs/TiO$_2$/Ni-Pi photoanode.

TiO$_2$/Ni-Pi photoanode film via cross-sectional focused ion beam (FIB) integrated high-resolution transmission electron microscopy (HR-TEM), scanning transmission electron microscopy (STEM), and energy dispersive spectroscopy (EDX), as shown in Fig. 3b–n. Here, we closely investigated the interfacial properties of GaAs/TiO$_2$ and TiO$_2$/Ni-Pi via high-resolution HR-TEM images, which showed the development of a heterogeneous structure. Figure 3 shows that the chemical etching process produces an intrinsic oxide layer with a constant thickness (~10 nm) between GaAs and TiO$_2$. These results are also confirmed by XPS, as shown in Supplementary Fig. 6a–f.

The generation of oxygen vacancies (O$_V$) verified by electron paramagnetic resonance (EPR) was recorded to survey the presence of O$_V$ or Ti$^{3+}$, with paramagnetic species containing unpaired electrons (Supplementary Fig. 7). It can be seen that the TiO$_2$ layer has a definite thickness of about ~30 nm, with considerable crystallinity, and is firmly attached to the Et-GaAs surface, where the compact film composed of the nanoporous TiO$_2$ particles on GaAs film was confirmed by FE-SEM images as shown in Supplementary Fig. 8. These results suggest that the interfacial crystalline TiO$_2$ protective film can be formed by a simple deposition method to boost the interfacial adhesion and surface area. In Fig. 3b, the deposited thickness of the co-catalyst is estimated to be ~10 nm, and is intimately attached to the TiO$_2$ interface. In particular, the dense electrodeposited amorphous Ni-Pi layer in close contact with the TiO$_2$ interface can efficiently trap and transport holes (h$^+$) to the electrolyte, facilitate a PEC reaction without affecting the light absorption ability of the GaAs film. In Fig. 3c, the well-resolved lattice lines (0.38 nm) in the bulk region are found to be matched with the highly crystalline (220) plane of orthorhombic GaAs phase. To further interpret the crystal properties at each interface, we plotted selected area electron diffraction of Et-GaAs/TiO$_2$/Ni-Pi, with the indicated areas (represented by squares 1, 2, 3, and 4 in Fig. 3b)

corresponding to the Et-GaAs bulk, Et-GaAs/native oxide interface, native oxide/TiO$_2$ interface, and TiO$_2$/Ni-Pi interface, respectively. By calculating the inverse lattice constant of GaAs, we matched the d-spacing in the [0001] and [10-10] directions (0.549 and 0.494 nm, respectively) with the single crystalline GaAs with (220) planes (Fig. 3d). The fast Fourier transform (FFT) pattern collected at the interface is a superimposed pattern from both native oxides of cubic As$_2$O$_3$, providing evidence of the overlapped FFT pattern of the mixed compounds (Fig. 3e). The distinct spots with the ring pattern of anatase TiO$_2$ (spot 3, Fig. 3f) correspond to the (004) plane, and the diffused ring pattern of amorphous Ni-Pi (spot 4, Fig. 3g) revealed the successful formation of Et-GaAs/TiO$_2$/Ni-Pi.

To delicately figure out the elemental distribution of each layer, we measured the EDX line profile by repeating the slight etching process from the top surface to the bulk direction, like the inset STEM image of Fig. 3h. The result is summarized in Fig. 3h. we certified the Ni-Pi thickness of ~10 nm, where Ni and Pi are uniformly and densely distributed on the surface, and once again confirmed the presence of the well-bonded amorphous Ni-Pi phase on the TiO$_2$ surface. Second, we found the exact thickness of the TiO$_2$ layer prepared by the facile spin-coating method to be ~30 nm. This is well accordance to the thickness of TiO$_2$ layer measured by alfa-step profiler, as shown in Supplementary Fig. 9. As expected, about ~10 nm of amorphous native oxide made during the chemical etching process was converted to a weakly crystalline oxide during post-annealing after TiO$_2$ coating in an N$_2$ atmosphere. Accordingly, the atomic concentration (%) of O approached ~100% at the initial position and gradually decreased with the scan position, steadily decreasing to ~20% at 50 nm, indicating the formation of an amorphous Ni-Pi bifunctional co-catalyst, TiO$_2$ passivation layer, and native oxides of the GaAs surface up to 50 nm in depth.

**Table 1 | Simulated values of each photoelectrode using the suggested equivalent circuit**

| Photoanode | Rs (Ω) | $R_{SC}$ (Ω) | $R_{CT}$ (Ω) | CPE1(Q1)nF | CPE2(Q2) nF | n1 | n2 |
|---|---|---|---|---|---|---|---|
| GaAs | 3 | 15.8k | 22.5k | 174.51 | 476.8 | 0.85 | 0.74 |
| Et-GaAs | 5 | 7.02k | - | 79.60 | - | 0.92 | - |
| Et-GaAs/TiO$_2$ | 4.5 | 3.70k | - | 68.72 | - | 0.94 | - |
| Et-GaAs/TiO$_2$/Ni-Pi | 4.7 | 1.47k | 0.95k | 42.2 | 147.5 | 0.88 | 0.65 |

Figure 3i−n corresponds to the cross-sectional views of the STEM image, integrating energy dispersive spectroscopy (EDX elemental mapping (Ga, As, Ti, O, Ni, and P) of the Et-GaAs/TiO$_2$/Ni-Pi film with the bright-field high-angle annular dark-field scanning transmission electron microscopy (HAADF-STEM) image (inset of Fig. 3h). We observed that Ga and As have an almost 1:1 ratio. This is further confirmed by the EDX spectrum, as shown in Supplementary Fig. 10. We can also see the EDX spectra of the other elements, such as O, Ni, and P. Here, the atomic concentration (%) of oxygen provides valuable information on the thickness of the passivation and co-catalyst layers, as shown in Fig. 3l.

Figure 4a shows the *J−V* behavior of GaAs, Et-GaAs, Et-GaAs/TiO$_2$, and Et-GaAs/TiO$_2$/Ni-Pi photoanode films in 1.0 M NaOH(aq) (pH = 13.6) in the intermittent illumination using the simulated air mass (AM) 1.5 G illumination (100 mW·cm$^{-2}$), to allow for a comparison of the LSV results and the dark currents. The inset of Fig. 4a shows the onset potential ($V_{on}$) to show −0.37 −0.32, −0.4, and −0.37 $V_{RHE}$ for GaAs, Et-GaAs, Et-GaAs/TiO$_2$, and Et-GaAs/TiO$_2$/Ni-Pi films, respectively. Etched GaAs exhibited a slight positive shift in $V_{on}$ caused by the native oxide layer, and conversely, the TiO$_2$-coated Et-GaAs film exhibited a negative shift in $V_{on}$. Electrochemical etching induced on the surface nanotexturing to enhance the surface reaction sites/light absorption significantly increased $J_{ph}$ to 25 mA·cm$^{-2}$. As the most interesting electrode, the Et-GaAs/TiO$_2$ electrode reached a saturated photocurrent density of ~24 mA·cm$^{-2}$ at 0.1 $V_{RHE}$, close to the theoretical value of 28 mA·cm$^{-2}$, with a clear shift in the $V_{on}$. Herein, the crystalline TiO$_2$ layer with the graded oxygen defects is used as the protective layer. The graded oxygen defects/Ti$^{3+}$ in the TiO$_2$ layer confirmed by EPR (Supplementary Fig. 7) and XPS (Supplementary Fig. 11) allow photogenerated carriers to flow easily to the surface[26,27]. Further, to avoid the loss of TiO$_2$−either by oxidation or a higher density of defects−and to increase the rate of the surface kinetics in the PEC reaction, we deposited the co-catalyst on the GaAs to capture and transport the photogenerated holes toward the surface to accelerate the PEC reaction rate at the surface interface[28]. Also, to achieve the complete PEC performance of intrinsic GaAs without a recombination event on the surface, the Et-GaAs/TiO$_2$ system needs to incorporate the benchmark earth-abundant bifunctional HER/OER electrocatalyst[29–33]. Here, the nickel-phosphate (Ni-Pi) co-catalyst is applied for efficient charge transfer into the electrolyte. In our study, the nickel phosphate (Ni-Pi) has been also shown to be a promising electrocatalyst with Et-GaAs/TiO$_2$ photoanode for water oxidation under alkaline conditions to promote the necessary multiple electron and proton transfer toward the electrolyte and sometimes, it can undergo reversible redox reactions, facilitating the transfer of electrons[34]. The shift of the onset potential to −0.37 $V_{RHE}$ indicates that the survived charges from the catalytically active sites were transferred to the electrolyte and participated in water oxidation, followed by a subsequent increase in photocurrent density of 23.52 mA·cm$^{-2}$ and 25.77 mA·cm$^{-2}$ at 0 and 1.23 $V_{RHE}$, respectively.

To figure out the quantitative $J_{ph}$ contribution between each modification, such as electrochemical etching and surface passivation, the incident photon-to-current efficiency (IPCE), defined as the ratio of the number of collected photogenerated electrons to the number of incident photons, was measured at 0 $V_{RHE}$ and is shown in Fig. 4b. In general, the IPCE value is described by the following Eq. (1), compared with the quantitative *J−V* curve (Fig. 4a)[35]:

$$IPCE = (1,240^* J)/(\lambda^* P_{light}) \tag{1}$$

where J is the measured photocurrent density (mA·cm$^{-2}$) at a particular wavelength, λ(nm) is the wavelength of incident light, and $P_{light}$ is the incident light (mW·cm$^{-2}$). The photoresponse exhibited a sharp absorption edge at 870 nm, which corresponds to the bandgap ($E_g$ = 1.4 eV) of GaAs. The optical reflection loss at the smooth GaAs electrode surfaces accounts for ~25% of the incident photons at each wavelength. Therefore, the Et-GaAs film exhibits enhanced IPCE values mainly due to suppression of the reflection from the front surface in all wavelength regions. In the 550−900 nm range, Et-GaAs/TiO$_2$/Ni-Pi achieved the greater IPCE value, which can be explained by the improved light absorption of Et-GaAs with TiO$_2$ and the excellent hole transfer toward the surface caused by the high defect concentration in crystalline TiO$_2$.

To further understand the photovoltage generation by addition of passivation layer and the recombination loss at the interface we measured the open-circuit voltage (OCV) decay curves (Fig. 4c) of GaAs, and Et-GaAs/TiO$_2$ films under illumination (OCV$_{light}$) and darkness (OCV$_{dark}$)[36]. Fig. 4c shows the OCV values to be 0.3, and 0.95 V for GaAs, and Et-GaAs/TiO$_2$ photoanodes, respectively. Higher OCV values generate a more favorable driving force for water oxidation, and a large band bending is beneficial for charge and photovoltage generation, as determined by the larger difference between the electron quasi-Fermi levels of GaAs and Et-GaAs/TiO$_2$, which resulted in effective charge separation under PEC water splitting.

Figure 4d shows the Nyquist plots of the pristine GaAs, Et-GaAs, Et-GaAs/TiO$_2$, and Et-GaAs/TiO$_2$/Ni-Pi films at OCV under the solar illumination condition to investigate the effects of the TiO$_2$ and co-catalyst layers in terms of the charge transfer phenomena. Nyquist plot with a single semicircle of Et-GaAs and Et-GaAs/TiO$_2$ films represents a simplified Randles circuit (inset of Fig. 4d) consisting of $R_S$, $R_{CT}$, and CPE connected in parallel with $R_{CT}$ and the fitted values of each sample are shown in Table 1. Rs is associated with the resistance of the electrolyte, current collector, and other components in the system. $R_{CT}$ represents the resistance associated with the charge transfer process at the electrode/electrolyte interface. The CPE is used instead of a pure capacitance in equivalent circuit models because it better represents the non-ideal capacitive behavior of the system. The Nyquist plot of two semicircles indicates the presence of two distinct time constants in GaAs and ET-GaAs/TiO$_2$ films. In this case, the modified Randles circuit can be used, which includes an additional parallel combination of resistance as well as capacitance (CPE1 and CPE2) being the constant phase elements with respective parameters of Q1, n1 and Q2, n2[37]. The corresponding bode plots and magnitude of the impedance verses logarithmic frequency shown in Supplementary Fig. 12. The resistors ($R_{SC}$, $R_{CT}$) in the equivalent circuit represent the different charge transfer resistances associated with different processes in the photoanode. These resistances can be elucidated by the recombination of photo-generated electron-hole pairs, the charge transfer kinetics at the semiconductor/electrolyte interface, and the transport of charge carriers through the bulk of the semiconductor. Smaller semicircles are observed in the photoelectrodes (in order: Et-GaAs/TiO$_2$/Ni-Pi > Et-GaAs/TiO$_2$ > Et-GaAs > GaAs) under illumination, which means that the

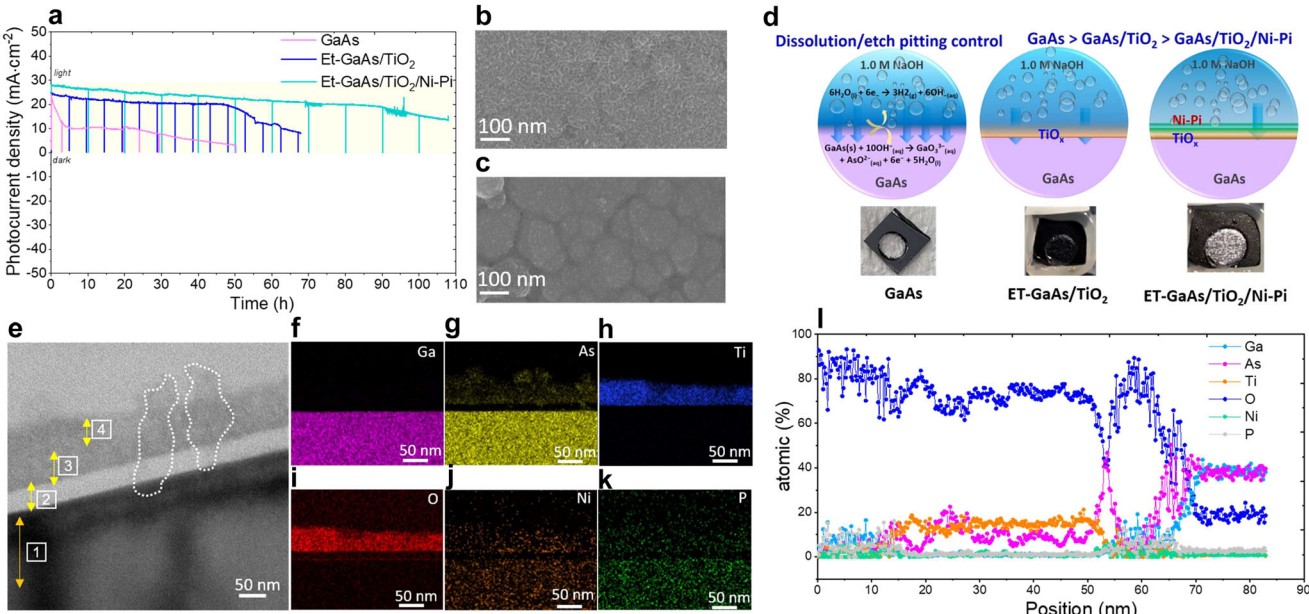

**Fig. 5 | PEC stability and degradation pathway control. a** Long-term stability test of Et-GaAs/TiO$_2$/Ni-Pi photoanode at 1.23 V$_{RHE}$ under 1-sun illumination. **b, c** Surface scanning electron microscopy images of Et-GaAs/TiO$_2$/Ni-Pi before and after stability testing. **d** Illustration representing the surface and interfacial dissolution phenomena happening during the PEC reaction of GaAs, Et-GaAs/TiO$_2$, and Et-GaAs/TiO$_2$. **e** Cross-sectional HR-TEM image showing the interfacial change of Et-GaAs/TiO$_2$/Ni-Pi after the stability test. The labels of 1, 2, 3, and 4 represent GaAs, native oxides, TiO$_2$ and Ni-Pi, respectively. The white lines indicate electrolyte penetration to reach the GaAs interface. **f–k** Elemental mapping images of Ga, As, Ti, O, Ni, and P of Et-GaAs/TiO$_2$/Ni-Pi after the stability test. **l** In-depth EDX line scanning of each element in Et-GaAs/TiO$_2$/Ni-Pi after the stability test, carried out in 1.23 V$_{RHE}$ under 1-sun illumination.

interfacial resistance of the Et-GaAs/TiO$_2$/Ni-Pi films is much smaller than that of bare GaAs. This finding leads to improved PEC performance, due to the advancement of the interfacial charge transfer rate after the addition of TiO$_2$ and Ni-Pi layers.

The effect of the Ni–Pi layer on Et-GaAs/TiO$_2$ film is again confirmed through PEC O$_2$ generation over time using a customized quartz cell, and is summarized in Fig. 4e. We constructed a PEC cell with argon gas bubbling over the surface of the material and used an artificial solar light (300-W Xe lamp) as a 1-sun illumination source. The O$_2$ yield was measured using GC/MS-YL6400 in in situ mode. Control experiments were conducted on the pristine GaAs photoanode. O$_2$ production can be observed at a low level on the bare sample; however, in the case of the Et-GaAs/TiO$_2$/Ni-Pi photoanode, the O$_2$ yield nearly doubled. The final O$_2$ yields over a 4 h period under these conditions were 200 and 415 µmol·cm$^{-2}$, in GaAs and Et-GaAs/TiO$_2$/Ni-Pi, respectively. The Et-GaAs/TiO$_2$/Ni-Pi photoanode revealed a remarkable improvement in the O$_2$ production rate with high photostability. Furthermore, the O$_2$ evolution rate corresponding the FE(%) of GaAs, Et-GaAs, Et-GaAs/TiO$_2$ and Et-GaAs/TiO$_2$/NiPi photoanode as a function of time was as shown in Supplementary Fig. 13. This is help to understand the side photocurrent due to the photo-corrosion effect in the photoanode film during water oxidation. These side reactions (e.g., non-conductive native oxide or non-faradic charging process at the interface etc.) induce to a side current that competes with the desired photocurrent, leading to lower FE in the GaAs and Et-GaAs. However, there is no meaning degradation observed in the photoelectrodes such as Et-GaAs/TiO$_2$, and Et-GaAs/TiO$_2$/NiPi photoanodes due to the surface passivation effect, surely demonstrating the stable PEC performance and in particular, co-catalyst boosts the rapid charge transfer reaction at the surface interface.

Figure 4f presents a schematic band alignment diagram of an Et-GaAs/TiO$_2$/Ni-Pi photoanode film in contact with the electrolyte under the working PEC conditions. Pure GaAs shows valence bands (VBs) and conduction bands (CBs) that are in good agreement with the water oxidation/reduction potentials. With the addition of crystalline TiO$_2$, we can see that the CB of TiO$_2$ is lower than the CB of GaAs, and the VB of TiO$_2$ is higher than that of GaAs. Therefore, the photogenerated electrons in GaAs are excited to the CB, and at the same time, holes in the VB are combined with electrons in the TiO$_2$ CB. In addition, the energetic holes in TiO$_2$ easily migrate to the Ni-Pi electron co-catalyst layer, which comprises hole storage, and are finally injected into the surface with high potential energy for water oxidation reaction.

Figure 5a shows the long-term stability of the GaAs, ET-GaAs/TiO$_2$, and ET-GaAs/TiO$_2$/Ni-Pi photoanodes at 1.23 V$_{RHE}$. Bare GaAs films showed a significant initial drop and continued to drop over time, maintaining 50% and 10% of the initial photocurrent density value after 5 h and 50 h, respectively, revealing the poor stability over time. Also, the substrate was completely damaged by the etching from top to bottom, as shown in the photograph in the inset of Fig. 5d. In contrast, the photocurrent density for the Et-GaAs/TiO$_2$ sample remained stable for 50 h and then gradually decreased to -10 mA over the next 20 h. The photocurrent densities were fairly sustainable between the initial time and 50 h, revealing excellent stability due to the conformal crystalline TiO$_2$ protective layer. Slow photocurrent decay after 50 h was observed in the Et-GaAs/TiO$_2$ photoanode in strong alkaline electrolyte, attributed to the exfoliation or dissolution of TiO$_2$ or the conversion to unstable intermediate oxides under the higher applied oxidation potential. We measured the surface topology of the Et-GaAs/TiO$_2$ photoanode after the long-term stability testing (Supplementary Fig. 14) via atomic force microscopy (AFM) analysis, revealing the lower damage of the surface nanostructure compared to bare GaAs. This confirmed that the simple spin-assisted TiO$_2$ layer significantly protected GaAs from immediate photocorrosion under the PEC working condition. Finally, to mitigate the aforementioned limitations of Et-GaAs/TiO$_2$ photoanodes, we electrodeposited bifunctional Ni-Pi co-catalysts on the Et-GaAs/TiO$_2$ layer. The Et-GaAs/TiO$_2$/Ni-Pi photoanode achieved remarkable stability (-95 h) while maintaining high photocurrent density (> 20 mA·cm$^{-2}$). Beyond 95 h, we observed slow decay, probably due to the formation of poor conductive NiO$_x$ from the oxidation reaction. The surface topology of the Et-GaAs/TiO$_2$/Ni-Pi

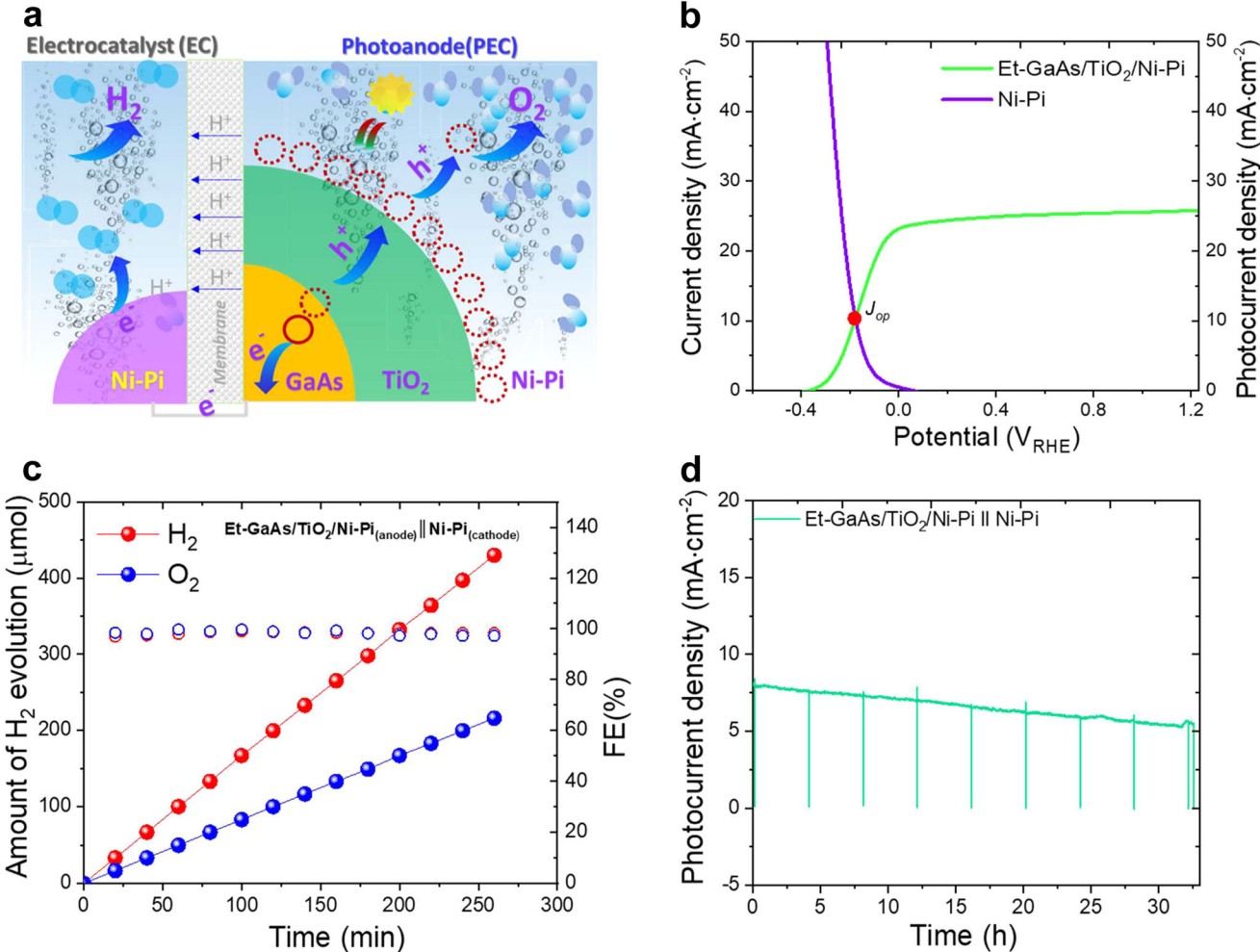

**Fig. 6 | Photoelectrode (PEC)–electrocatalyst (EC) tandem cell performance and stability under 1-sun illumination. a** Simple schematic model of unassisted (zero-bias) tandem photoelectrode -electrocatalyst film (PEC-EC) device configuration for solar water splitting. **b** Merged LSV curves of Et-GaAs/TiO₂/Ni-Pi photoanode and Ni-Pi electrocatalyst under alkali condition. **c** The H₂ and O₂ evolution rate with Faradic efficiency (%) under tandem configuration and, (**d**) Stability of unassisted zero-bias PEC-EC tandem cell under illumination and darkness at constant time intervals in 1 M NaOH. The inset of (**d**) represents the configuration of the tandem cell, including two electrodes composed of anode and cathode compartments and equally separated by an anion exchange membrane. The gas bubbles in the anode and cathode sides are O₂ and H₂ gases, respectively.

photoanode after long-term stability testing was also measured via AFM analysis (Figure S14), revealing a well-maintained surface topology, even in the severe PEC testing condition. To confirm reliable and durable operation under the lower intensity of illumination, the PEC stability of Et-GaAs/TiO₂/Ni-Pi was tested at 1.23 $V_{RHE}$ under 0.5-sun illumination (Supplementary Fig. 15), with an initial photocurrent of 15 mA·cm⁻². Even though the photocurrent density is half of a 1-sun illuminated cell, the stability of photoanode film extends over 200 h without any serious degradation. Low and high perturbations of the photocurrent density come from electrolyte evaporation during the long-term PEC testing. Using a lower intensity of illumination can increase PEC stability, but lower photovoltage and photocurrent density are inevitable. Furthermore, the surface morphologies before and after long-term stability testing were compared and shown in Fig. 5b, c, revealing that the Ni-Pi layer remains, but the interfaces between the Ni-Pi nanostructures are separated to make a bundled form. In addition, for a more in-depth understanding of the degradation process during the PEC reaction, we analyzed the electrolyte collected during the stability test to detect the dissolved species by inductively coupled plasma-optical emission spectrometry(ICP-OES). The results are summarized in Supplementary Fig. 16. Severe dissolution happened in the bare GaAs film, whereas the compact and uniform

TiO₂ layer in the Et-GaAs/TiO₂ film strongly controlled the dissolution rate of the GaAs film, slowing the diffusion of electrolyte toward the GaAs surface.

Figure 5d supports the detailed mechanism scheme of the expected dissolution rate in GaAs, GaAs/TiO₂, and GaAs/TiO₂/Ni-Pi photoanode films. Under the strong alkali solution, the dissociation rate of GaAs is fast where 6e⁻ transfer is involved to dissociate the Ga-As bonding. Therefore, the native oxides are easily generated during the dissolution process of the GaAs film. Meanwhile, thin TiO₂ is a good medium to avoid direct contact with the electrolyte on the GaAs surface and to accompany the water oxidation reaction under no influence on the pristine GaAs film. This led to the good stability associated with the GaAs/TiO₂ film. However, some pinholes slowly appeared in the thin TiO₂ layer after a certain length of PEC testing, allowing electrolyte diffusion through the pinhole regions toward the GaAs surface, and finally causing minor degradation before taking out the film against the substrate mechanically. Finally, the selective benchmark co-catalyst is needed to avoid this slight degradation, even after the passivation layer. Ni-Pi is a dual function electrocatalyst for overall water splitting and is suitable for a wide pH range. Ni-Pi co-catalysts are electrodeposited on GaAs/TiO₂ film to avoid interfacial e⁻/h⁺ recombination, promoting the rapid extraction of photogenerated holes,

which are subsequently transferred to the solar water oxidation. In Supplementary Figs. 17, 18, the Ni-Pi exhibits excellent bifunctional activity for water electrolysis, a process that involves the simultaneous HER and OER. Nickel provides active sites for the electrocatalytic reactions, while phosphate ions stabilize the electrocatalyst and prevent agglomeration or dissolution of the active sites. This makes them a promising candidate for the development of efficient and cost-effective electrocatalysts on the photoanode film[34,38].

For detailed surface and interface understanding for the photoanode stability, the microstructure, interfacial thickness, and spatial component distribution of Et-GaAs/TiO$_2$/Ni-Pi films after stability testing were investigated with XPS (Supplementary Fig. 19a–f), cross-sectional focused ion beam (FIB) integrated high-resolution transmission electron microscopy (HR-TEM), scanning transmission electron microscopy (STEM), and energy dispersive spectroscopy (EDX), arranged in Fig. 5e–k. The marked areas in Fig. 5e show the changes in the surface and interfacial regions that occurred during PEC stability testing. A small, leaky surface was created under the harsh analytical conditions, allowing the electrolyte penetration to reach the GaAs interface, which was directly oxidized to create an oxide layer that grew toward the surface (indicated by the white dashed line). Figure 5f–k displays the elemental distribution of Ga, As, Ti, O, Ni, and P elements in the ET-GaAs/TiO$_2$/Ni-Pi photoanode, revealing the interface and surface transformation after PEC testing. Although there is no significant change in the Ga distribution, a large change is observed in the distribution of As element, due to the fast dissolution rate to the electrolyte. The As arrived at the surface to form a native oxide, and it matched with the XPS results in Supplementary Fig. 19. The concentrations of Ti and O did not change, demonstrating no apparent damage during stability testing. A slight loss of Ni was observed, compared to the concentration of P, as PO$_4^{3-}$ ions are quite stable in a strong alkali environment. Since NiPi can form a thin protective layer typically composed of nickel hydroxide (Ni(OH)$_2$) or nickel oxyhydroxide species (NiOOH) on its surface when exposed to alkali electrolytes. This passivation layer acts as a barrier, preventing further interaction between the electrolyte and the nickel phosphate compound, which enhances its stability. This nickel hydroxide or nickel oxyhydroxide species can participate in reversible (Ni$^{2+}$ to Ni$^{3+}$) redox reactions, providing the additionally active co-catalyst material for the photoanode film. When this layer is damaged or partially removed, it can spontaneously reform due to the electrochemical reactions at the interface between the nickel phosphate and abundant OH$^-$ containing alkali electrolyte[38]. We further studied the elemental distribution in the GaAs/TiO$_2$/Ni-Pi photoanode by a STEM-EDX line-scanning profile, as shown in Fig. 5l. The surface was rich in O, the core was rich in Ga and As, and Ti, Ni, and P were distributed in selected regions of the film. The Ni-Pi thickness was estimated to be about 15 nm from the surface, and Ga was also found on the surface, signifying the formation of a 10 nm-thick Ga-O-P phase with increasing P concentration. The As element found at 10 nm increased gradually with depth, finally reaching a constant atomic percentage around 80 nm. Similarly, a constant 30 nm-thick TiO$_2$ layer was observed in the GaAs/TiO$_2$/Ni-Pi photoanode film, like in the initial film, indicating no evidence of TiO$_2$ loss. A Ni-Pi phase with a Ga-O phase was observed at the depth of 55–65 nm, indicating that the pinhole may induce the electrolyte penetration toward the bulk GaAs. Remarkably, a mixed elemental state in the form of Ni-Pi-O-Ga can therefore be observed. After reaching a depth of 70 nm, a rapid decrease in O concentration and an increase in Ga and As concentrations are together observed in the bulk film, and afterward, there is no further change in the film.

Finally, the surface- and interface-modified Et-GaAs/TiO$_2$/Ni-Pi photoanode film is integrated with the PEC-EC tandem cell configuration with the bifunctional Ni-Pi electrocatalyst, as described in Fig. 6a[39]. In the PEC-EC configuration, the photoanode and cathode are separated into their respective compartments through an anion

exchange membrane. To get sufficient photovoltage two Et-GaAs/TiO$_2$/Ni-Pi photoanode are connected in series. The Et-GaAs/TiO$_2$/Ni-Pi photoanode film is illuminated under 1-sun condition, and the photogenerated e$^-$/h$^+$ are separated toward the cathode and anode surfaces to generate H$_2$ and O$_2$, respectively. In order to attain a STH conversion efficiency under zero bias, the photoanode and cathode should have sufficient potential with photo(current) density. Figure 6b shows the merged LSV of an Et-GaAs/TiO$_2$/Ni-Pi photoanode and Ni-Pi electrocatalyst obtained from 1 M NaOH in a three-electrode configuration and the operating photocurrent density (J$_{op}$) of ~10.2 mA·cm$^{-2}$ at zero bias, which the curve intersects.

Supplementary Fig. 20 shows the LSV curve of Et-GaAs/TiO$_2$/Ni-Pi‖Ni-Pi measured in the tandem cell configuration under chopped illumination in 1 M NaOH. This Et-GaAs/TiO$_2$/Ni-Pi‖Ni-Pi tandem configuration results in the best unassisted bias-free water splitting device with the highest J$_{ph}$ of ~7.6 mA·cm$^{-2}$ and an STH efficiency of 9.5%. The variation in photocurrent between a 3-electrode and 2-electrode configuration is attributed to the energy loss due to the membrane used in H-type cells. The gas bubbles in the anode and cathode side are O$_2$ and H$_2$ gases and are evaluated in the tandem configuration (Fig. 6c). The corresponding H$_2$ and O$_2$ gases were produced at a 2:1 stoichiometric ratio, and their amounts increased linearly with time. Meanwhile, this PEC cell exhibits a FE close to 100% for overall water splitting, implying that essentially all photogenerated charge carriers are utilized to generate gases. In addition, the stability of this tandem cell was tested in unbiased light and dark conditions, as shown in Fig. 6e. During the water splitting reaction, the very stable photocurrent densities were observed over 10 h of illumination with vigorous gas bubbles through both the photoanode and cathode compartments, confirmed in the inset of Fig. 6d.

In conclusion, this work demonstrates the possibility of overcoming key challenges to realize a highly durable PEC cell based on bare GaAs with severe photodegradation under standard PEC working conditions. A thin layer of TiO$_2$ with a thickness of ~30 nm is used for the surface passivation effect, inhibiting electrolyte penetration and minimizing recombination losses at the photoanode/electrolyte interface. Finally, the PEC exhibited very efficient and stable performance for 50 h while minimizing photocurrent losses through fast charge transport rates in the oxygen-deficient TiO$_2$ layer. The Et-GaAs/TiO$_2$/Ni-Pi photoanode exhibits a remarkable photocurrent density of 25 mA·cm$^{-2}$ at 1.23 V$_{RHE}$. Moreover, the TiO$_2$/Ni-Pi effectively inhibits photocorrosion of the Et-GaAs and enables long-term operation for >100 h with minimal photocurrent loss. Furthermore, a standalone unbiased PEC tandem device comprising an Et-GaAs/TiO$_2$/Ni-Pi‖Ni-Pi photoanode and cathode can achieve a record STH conversion efficiency of 9.5%, representing the most efficient PEC water splitting device to date.

## Methods
### Materials and chemicals
A *n*-type Si-doped commercial single-side polished GaAs <100> wafer (carrier concentration: 0.05−2 × 10$^{18}$ cm$^{-3}$; thickness: 460 μm; diameter: 50.8 mm) was used as a substrate.

### Preparation of substrate
GaAs substrate was soaked in 10% HCl solution for 5 min to remove surface contaminants, then rinsed with deionized (DI) water, flashed with a stream of N$_2$(g), and stored in a vacuum package.

### Etched GaAs photoanode
An electrochemical etching method was adopted to create a porous GaAs surface. The GaAs substrate was cut into 1.5 × 1.5 cm$^2$ pieces and dipped in 50 mL of 2 M NaCl in a two-electrode electrochemical configuration composed of GaAs and Pt foil as working and counter electrodes, respectively, and a constant current of −5 mA was applied

for 10, 20, and 30 min. The prepared samples were denoted as Et-GaAs-10 min, Et -GaAs-20 min, and Et-GaAs-30 min, respectively.

## TiO$_2$ passivation on Et-GaAs photoanode

A thin TiO$_2$ layer was then spun over the etched GaAs photoanode surface as the protecting layer. Herein, 5 mM of titanium butoxide (Ti(OBu)$_4$) precursor dissolved in ethanol was stirred for 12 h at room temperature. In order to achieve the uniform coating, the controlled parameters such as precursor volume, spin speed, spin speed, etc., were used. Essentially, the parameters used here were 20 μL volume, 1000 rpm speed/10 s, followed by 2000 rpm/30 s. This slightly deposited process was repeated multiple times to achieve a uniform thickness of TiO$_2$. Finally, the samples were annealed at 300 °C under the N$_2$ atmosphere for 30 min. The fabricated samples are referred to as Et-GaAs/TiO$_2$.

## Ni-Pi deposition on Et-GaAs/TiO$_2$ photoanode

For the Ni-Pi electrodeposition, a solution consisting of 100 mM nickel sulfate hexahydrate (NiSO$_4$·6H$_2$O), 300 mM sodium hypophosphite (NaH$_2$PO$_2$), and 300 mM ethylenediamine was prepared. The electrodynamic deposition was carried out in a three-electrode cell comprised of Et-GaAs/TiO$_2$ photoanode as the working electrode, Pt foil as the counter electrode, and sat. Ag/AgCl as reference electrode by cyclic voltammetry at a scan rate of 5 mV·s$^{-1}$ for 25 cycles within a voltage range of −1.5 to 0.2 V versus reversible hydrogen electrode (RHE, briefly abbreviated as V$_{RHE}$). The as-prepared Et-GaAs/TiO$_2$ was cleaned with ethanol and DI water and then dried in N$_2$ gas. The fabricated samples are referred as Et-GaAs/TiO$_2$/Ni-Pi.

## Preparation of Bi-functional electrocatalyst for HER and OER study

Ni-Pi was deposited using the electrochemical deposition method on nickel foam (NF). The substrate (1 × 1 cm$^2$) was immersed in an aqueous solution containing 0.2 M nickel sulfate hexahydrate (NiSO$_4$·6H$_2$O), 0.5 M sodium hypophosphite (NaH$_2$PO$_2$), and 0.5 M ethylenediamine at room temperature for 100 cyclic voltammetry cycles in the range of −1.5 to 0.2 V$_{RHE}$. For further comparison, M-OH (NiOOH) was also fabricated using a similar method, and the HER and OER results were compared with the Ni-Pi catalyst.

## Physical characterization

The structural and crystalline properties of the GaAs, Et-GaAs, Et-GaAs/TiO$_2$, and Et-GaAs/TiO$_2$/Ni-Pi photoanode films were studied by a high-resolution X-ray diffraction (HR-XRD, PANalytical, X'Pert PRO) instrument (40 kV and 30 mA) as well as focused ion beam (FIB) integrated high-resolution transmission electron microscopy (HR-TEM, JEOL-3010 instrument, 300 kV). The film morphology was examined by Field Emission−Scanning Electron Microscopy (FE−SEM) with an S4800 instrument (Hitachi Inc.) at 20 kV acceleration voltage. The UV-Vis absorption spectra, in the range of 400–1000 nm, were measured by PerkinElmer UV−Vis Lambda 365 spectrometry. The chemical composition and chemical states were analyzed by X-ray photoelectron spectroscopy (XPS) with monochromatic Al $K\alpha$ irradiation. In addition, high-angle annular dark-field (HAADF) imaging and energy dispersive X-ray analysis (EDX) elemental mapping were used to show three-dimensional morphological features and the distributions of each element.

## PEC measurements

The working electrodes used in this study were GaAs, Et-GaAs, Et-GaAs/TiO$_2$, and Et-GaAs/TiO$_2$/Ni-Pi photoanodes, while the counter electrode was a Pt plate (1 × 2 cm$^2$), and the reference electrode was Hg/HgO (filled with 1 M NaOH) electrode (0.14 V compared to a normal hydrogen electrode). The potential of the reference electrode can be converted to RHE ( = NHE at pH = 0) using Eq. (2):

$$E_{RHE} = E_{Ag/AgCl} + 0.0591V \cdot pH + 0.14V \qquad (2)$$

The PEC test was conducted using an aqueous electrolyte containing 1 M NaOH (pH = 13.5). An Oriel Newport 150-W Xenon solar light simulator (100 mW·cm$^{-2}$, AM 1.5 G) was used as the light source. In all cases, the illuminated area was 0.2 cm$^2$. In order to measure $J–V$ curves, potential was swept in the positive direction at 20 mV·s$^{-1}$ under constant illumination and chopped light (alternating between dark and light every 2 s). At a potential bias of 1.23 V$_{RHE}$, incident photon-to-current conversion efficiency (IPCE) was measured from 300 nm to 1000 nm. To produce the monochromatic beam, a 150-W Xenon lamp was used. Si photodiodes certified by NREL were used in the calibration of 1-sun illumination. The cell resistance was estimated by electrochemical impedance spectroscopy (EIS). An impedance spectrometer (Nova) equipped with a potentiostat (Autolab/PGSTAT, 128 N) was used for the EIS measurement.

Gas chromatography (GC, in situ operation mode) was used to analyze the headspace of an airtight cell to measure H$_2$/O$_2$. To instantly flush away O$_2$ from the working electrode, Argon (Ar) gas was pumped through the cell during the measurement. As a precaution against unexpected reactions, we made an indium ohmic contact along the edges of the samples and coated them with nonconductive epoxy resin. This was to ensure only the samples came into contact with the electrolyte during PEC. About 1.0 cm$^2$ of the sample area was in direct contact with the electrolyte. A carrier gas, Ar, of 20 mL per minute, was used for the gas chromatography. A metal line was used to take on-line gas from the headspace of the cell into the gas-sampling loop of the GC every 20 min to determine the amount of H$_2$ and O$_2$. The faradaic efficiency (FE) was calculated using Eq. 3:

$$FE(\%) = \frac{z \times n \times F}{Q} \times 100 \qquad (3)$$

where z is the evolved O$_2$ (mol) gas, $n$ is the electron involved in the reaction ($n$ = 2 and 4 for H$_2$ and O$_2$, respectively), F is the faradaic constant (96,485 C·mol$^{-1}$), and Q is the total charge passing through the working electrode[40].

## Data availability

The data that support the findings of this study are provided in the main text and the supplementary Information. Source data are provided with this paper.

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

## Acknowledgements

This research was supported by grants (2018R1A6A1A03024334, 2019R1A2C1007637, 2021M3I3A1082880, 2021R1I1A1A01044174) of the Basic Science Research Program through the National Research Foundation of Korea (NRF) (S.H.K.). The work was partially supported by the U.S. Department of Energy under Contract No. DE-AC36-08GO28308 with Alliance for Sustainable Energy, Limited Liability Company (LLC), the Manager and Operator of the National Renewable Energy Laboratory (K.Z.). The views expressed in the article do not necessarily represent the views of the DOE or the U.S. Government.

## Author contributions

M.A. conceived the ideas and designed the experiments, analyzed the data, and wrote the manuscript. R.S.K performed some material characterization. K.Z. took part in discussing the data and revising the manuscript. S.H.K. supervised the work and reviewed and revised the paper. All the authors discussed the results and reviewed the manuscript.

## Competing interests

The authors declare no competing interests.
