## [Peer Review File · Nature Communications]

REVIEWER COMMENTS

Reviewer #1 (Remarks to the Author):

Authors show a n-GaAs photoanode, reliable (30ks) and performance (25mA/cm²). This is achieved through optimization of the single parts. I have several questions:

1. The TiO₂ is 30 nm; the authors etch the GaAs heavily. What is the process to achieve uniformity?
2. Some figures in 3 of the nanoparticles seem drawn; please provide better figure (SEM). Others are OK.
3. F. 4-The use of TiO₂-NiPi should be weakness; Pi dissolve relatively easily.
4. Is it possible to perform an etch with nps while monitoring OCV and voltage.

Clearly this system have been well optimized.

Reviewer #2 (Remarks to the Author):

The authors reported chemically-etched GaAs decorated with TiO₂ and nickelphosphate (Ni-Pi) co-catalyst for photoelectrical water oxidation. There are following concerns which should be addressed before consideration of publication.

- (1) What is the depth of the nanoporous layer of the Et-GaAs samples?
- (2) The etching time should be optimized according to the overall PEC performance. What is the optimal etching time of the Et-GaAs for maximizing the PEC performance (highest photocurrent and smallest onset potential)?
- (3) The Nyquist plots and corresponding bode plots of GaAs, Et-GaAs, Et-GaAs/TiO₂, and Et-GaAs/TiO₂/NiPi photoanodes should be given together with all simulated parameters, which needs to be clearly explained.
- (4) What is the optimal thickness of TiO₂ for reaching the maximum photocurrent density and the stability? Does the TiO₂ layer coated on the Et-GaAs conformally or not? It is hard to see it from the elemental mapping results.
- (5) What is the faradaic efficiencies of the GaAs, Et-GaAs, Et-GaAs/TiO₂, and Et-GaAs/TiO₂/NiPi photoanodes? Is there a side photocurrent due to the photo-corrosion effect?
- (6) The photovoltage was measured to be 0.95 V. However, for a self-driven water splitting in the 2-electrode configuration without external bias, a minimum photovoltage of 1.23 V+ overpotentials to ensure fast kinetic reactions (~0.4 V) is required. With such a small photovoltage of 0.95 V, how can the Et-GaAs/TiO₂/Ni-Pi//Ni-Pi tandem cell drive the unassisted bias-free water splitting with J_{ph} of ~10.5 mA·cm⁻² and the solar-to-hydrogen conversion efficiency of 10.2%?

Reviewer #3 (Remarks to the Author):

In the present manuscript, the authors study a GaAs/TiO₂/Ni-Pi photoanode for water oxidation under alkaline conditions. Along this study, the authors performed a detailed structural, morphological, chemical and photoelectrochemical characterization of the employed materials through a wide range of different techniques. Additionally, a novel and functional unbiased tandem cell is proposed. The manuscript is well-written and the experiments well performed.

However, even though the manuscript is of interest to the solar fuels community and is well aligned with the scope of Nature Communications, there are some points that have to be addressed before publication in order to achieve the high-quality standards required in this prestigious journal.

1. In Figure 2c, the authors show the EIS results from bare and etched-GaAs samples. The authors should clarify at which potential are these Nyquist plots acquired and clarify why they proposed this equivalent circuit. Is the GaAs sample compact? Or is it possible that some Si substrate can be exposed to the electrolyte?

2. In Figure 4f, the authors show the band energy diagram of the complete photoanode. First of all, how were all these positions determined? On the other hand, how is possible that a 30 nm TiO₂ layer is not blocking the charge transfer from GaAs to the Ni-Pi co-catalysts with such band alignment? The authors must clarify this issue. If the role of the TiO₂ layer is passivation and protection, the thickness should be much lower, especially with such thermodynamic impediment. The authors have to clarify this point deeply. It is crucial.

3. Also, why do they deposit the TiO₂ layer over the intrinsic passivating layer of 10 nm that appears after the etching process?

4. In figure 4c, why do the authors propose the same equivalent circuit? Do no other features appear in the Nyquists' plots? At which potential were these plots acquired?

Reviewer #4 (Remarks to the Author):

This paper reported the use of Ni-P/TiO₂ deposited GaAs photoanodes to construct a PEC cell enabling overall solar water splitting. Stable, bias-free PEC solar water splitting with the STH efficiency of 10.2 % can be achieved. Although the topic of the work was significant with regard to the development of PEC solar water splitting, the current study failed to highlight the merits of the developed systems. The solution to the most critical issue facing PEC water oxidation was also missed. Considering the high criterion of the journal, I recommend to reject the manuscript of the present form. The following were specific comments.

Comments:

(1) Similar approaches to passivating GaAs photoanodes for enhancing the efficiency of PEC water oxidation have been widely reported (see examples at DOI:10.1126/science.1251428; DOI: 10.1021/acseenergylett.0c02521). The authors failed to highlight the difference of the present work from those already reported, making it difficult to appreciate the merits of the present system.

(2) Similar to the electrolytic OER, PEC water oxidation in acid electrolyte also imposes a much great challenge to limit the large-scale implementation of the economically viable photoanodes. The authors should also examine the practice of the current photoanode in acid electrolyte.

(3) In Fig 1D, the increase in absorbance for Et-GaAs across 400 to 900 nm region was accompanied by the increase in baseline intensity. This outcome suggested that significant light scattering occurred for Et-GaAs, which should not be confounded with the increase in photo-absorption.

(4) A table summarizing the current advancement of the-state-of-the-art GaAs-based PEC systems and tandem cells ever reported should be provided to enable a global performance comparison.

Response to Reviewers' Comments

First of all, we would like to thank the reviewers for their valuable time and attention in the review of our manuscript entitled “*Reliable bi-functional Ni-Pi/TiO₂ integration enables stable n-GaAs photoanode for water oxidation under alkaline condition*”. All reviewers have raised very valuable comments. We have addressed reviewers' comments point-by-point and have modified our manuscript accordingly. The revisions are highlighted in yellow in the revised manuscript. Implementing these changes strengthens this manuscript significantly. We are very grateful to the reviewers for their valuable time and effort in providing constructive comments.

Reviewer #1 (Remarks to the Author):

Authors show a n-GaAs photoanode, reliable (30ks) and performance (25 mA/cm²). This is achieved through optimization of the single parts. I have several questions:

Response: We thank the reviewer for the overall positive evaluation of this work. We greatly appreciate the reviewer's constructive comments to help improve the quality of our manuscript.

1. The TiO₂ is 30 nm; the authors etch the GaAs heavily. What is the process to achieve uniformity?

Response: We appreciate your feedback and value your opinion. We provide here an analysis and discussion of the fabrication process and outcome.

- Thickness of TiO₂ is $\sim 30 \pm 5$ nm by the cross-sectional HR-TEM images as mentioned in the manuscript (**Figure 3b**). The electrochemical etching of GaAs is not severe and does not cause serious surface damage. In **Figure R1(a-c)**, the corresponding morphology is shown after 10, 20, and 30 minutes of etching. Here, it is evident that the GaAs surface has been etched uniformly, to enable the uniformity of upper passivation layer.
- Moreover, α -step profilometer measurement is used to confirm the etching depth of GaAs (**Figure R2**). Alfa-step profiler can measure the thickness of thin films with a high degree of accuracy, typically on the order of nanometers. The recorded profile is typically used to determine the step height between the substrate and the film. This height difference corresponds to the thickness of thin film. According to **Figure R2**, the electrochemical etching time from 10 min to 30 min led to increase the etching depth from ~ 15 to 60 nm in the regular pattern. This can suggest that the etching time controls the etching depth of the GaAs.
- In order to achieve the uniform coating, the controlled parameters such as precursor volume, spin speed, etc., were used. Essentially, the parameters used here were 20 μ L volume, 1000 rpm speed/10s, followed by 2000 rpm/30s. This mild deposition process was repeated multiple times to achieve a uniform thickness of TiO₂.
- The above discussion and related figures (**Figures S1 and S2**) are updated in the revised manuscript.

Figure R1. Surface FE-SEM views of etched GaAs films (a) 10 min, (b) 20 min, and (c) 30 min.

Figure R2. Alfa-step depth profiles of the etched GaAs photoanode film with the etched time.

2. Some figures in 3 of the nanoparticles seem drawn; please provide better figure (SEM). Others are OK.

Response: We appreciate your feedback and value your opinion. To understand the surface morphology of the Et-GaAs/TiO₂ film, surface FE-SEM views were re-measured, showing the uniformly deposited TiO₂ layer on Et-GaAs film (**Figure R3** below). As suggested by the reviewer, we have added these SEM images in **Figure S8** in the revised manuscript.

Figure R3. (a-b) Surface FE-SEM views of Et-GaAs/TiO₂ film.

3. Figure 4, the use of TiO₂-NiPi should be weakness; Pi dissolve relatively easily.

Response: Thank you for the valuable comment on this point. The stability of NiPi layer is important because PEC water oxidation reaction requires co-catalysts that should be stable over long-time operation under harsh conditions, such as alkaline environments and light source. In our work, the nickel phosphate (NiPi) co-catalyst is prepared by electrochemical deposition, guaranteeing that the catalyst can be strongly adhered to the Et-GaAs/TiO₂ surface, and it can remain mechanically stable during PEC operation. Some validation points are given below:

- Metal phosphates have been found to be effective in promoting the water oxidation reaction by providing active sites for oxygen evolution and facilitating the transfer of photogenerated charges from the photoanode to the electrolyte. In our study, the nickel phosphate has been also shown to be a promising catalyst with Et-GaAs/TiO₂ photoanode for water oxidation under alkaline conditions because it is able to facilitate the necessary multiple electron and proton transfers and it can undergo reversible redox reactions that facilitate the transfer of electrons during the water oxidation reaction.
- Alkali environments typically have high pH values, and many compounds can undergo dissolution or degradation under such conditions. Interestingly, the nickel phosphate forms a stable structure due to the ionic bonding between the positively charged nickel ions (Ni²⁺) and the negatively charged phosphate ions (PO₄³⁻) as well as the covalent bonding within the phosphate ions themselves (P-O bonds). This robust lattice structure makes it less susceptible to dissolution or degradation in an alkali environment.
- Further, in alkali environments, phosphate ions (PO₄³⁻) can form stable complexes with alkali metal ions (Na⁺), where the hydroxide ions (OH⁻) react with the phosphate ions to form species such as HPO₄²⁻ and H₂PO₄⁻, respectively, as shown in equation (1). These complexes can reduce the concentration of the free phosphate ions available in the electrolyte, decreasing the solubility of nickel phosphate (Ni₃(PO₄)₂). As a result, the overall solubility of nickel phosphate in the alkali environment decreases, which leads to enhanced stability of the Et-GaAs/TiO₂/Ni-Pi photoanode.

- Nickel phosphate has a low solubility product (K_{sp}) value of ~ 4.74 x 10⁻³², indicating that it is thermodynamically stable and less likely to dissolve in in alkaline electrolytes. Since the K_{sp}

value is low, the equilibrium position lies far to the left, indicating that the compound prefers to stay in the solid phase rather than dissolve in water. In the case of nickel phosphate, the equilibrium reaction can be represented as equation (2):

- At a certain condition, the $\text{Ni}_3(\text{PO}_4)_2$ on the Et-GaAs/TiO₂ can be partially dissolved during the long-term stability test under harsh conditions. Since, the NiPi can form a thin protective layer typically composed of nickel hydroxide ($\text{Ni}(\text{OH})_2$) or nickel oxyhydroxide species (NiOOH) on its surface when exposed to alkali electrolytes. This passivation layer acts as a barrier, preventing further interaction between the electrolyte and the nickel phosphate compound, which enhances its stability. This nickel hydroxide or nickel oxyhydroxide species can participate in reversible (Ni^{2+} to Ni^{3+}) redox reactions, providing additionally active co-catalyst material for the photoanode film. When these layer is damaged or partially removed, it can spontaneously reform due to the electrochemical reactions at the interface between the nickel phosphate and abundant OH^- containing alkali electrolyte. This new co-catalyst formation further decreases the solubility of the unstable NiPO_4 leaving to the electrolyte.
- To confirm the presence of $\text{Ni}(\text{OH})_2$ or NiOOH on the Ni-Pi, we conduct XPS analysis using after stability tested Et-GaAs/TiO₂/Ni-Pi film. We found the presence of NiOOH species in the core-level $\text{Ni}2p$ spectra. For nickel oxyhydroxide (NiOOH), the $\text{Ni}2p_{3/2}$ peak is found at a slightly higher binding energy, around 856.09 eV, and the $\text{Ni}2p_{1/2}$ peak is observed around 873.61 eV, as shown in **Figure R4**. The formation of nickel oxyhydroxide (NiOOH) species is associated with the redox reaction between Ni^{2+} and Ni^{3+} ions. The redox reaction typically occurs at the surface of the Et-GaAs/TiO₂/Ni-Pi during the PEC stability test, following the equation (3):

- Additionally, the redox reaction between Ni^{2+} and Ni^{3+} in the formation of NiOOH can also contribute to the self-healing properties, as discussed earlier. The presence of the $\text{Ni}^{2+}/\text{Ni}^{3+}$ redox couple can facilitate better electronic conductivity in the Et-GaAs/TiO₂/Ni-Pi film by improving charge transfer kinetics at photoanode/electrolyte interface. Also, the formation of more active sites on the electrocatalyst surface results in faster reaction rates and lower overpotentials for increasing the energy conversion efficiency of the PEC device.
- As a conclusion, NiPi in the Et-GaAs/TiO₂ photoanode film is strong and significant. The above-discussed points are updated in the manuscript.

Figure R4. Core-level XPS spectra of (a) Ni2p, (b) P2p, and (c) O1s of after stability test of Et-GaAs/TiO₂/Ni-Pi films.

4. Is it possible to perform an etch with nps while monitoring OCV and voltage.

Clearly this system have been well optimized.

Response: We appreciate your feedback and value your opinion. According to the reviewer's suggestion, we monitored voltage and open-circuit voltage conditions during the etching process (**Figure R5**). Since the initial time, the voltage has been increasing continuously whereas, the OCV condition has not changed. This indicates that the etching process is stable and that the voltage and OCV conditions are not impacting substrate damage. Overall, this suggests that the etching process is progressing as expected.

Figure R5. Electrochemical etching process of the GaAs film.

Reviewer #2 (Remarks to the Author):

The authors reported chemically-etched GaAs decorated with TiO₂ and nickelphosphate (Ni-Pi) co-catalyst for photoelectrical water oxidation. There are following concerns which should be addressed before consideration of publication.

Response: We thank the reviewer for the overall positive evaluation of this work. We greatly appreciate the reviewer's constructive comments to help improve the quality of our manuscript.

1. What is the depth of the nanoporous layer of the Et-GaAs samples?

The alpha-step profilometer with a high degree of accuracy, typically on the order of nanometers is used to confirm the etching depth of GaAs (**Figure R6**). The recorded profile is typically used to determine the step height between the substrate and the film. This height difference corresponds to the thickness of thin film. According to **Figure R6**, the electrochemical etching time from 10 min to 30 min contributed to an increase in etching depth from ~ 15 to 60 nm. This shows that the etching time can be used to modulate the etching depth of the material. The above discussion and related figure (**Figure S2**) is updated in the manuscript.

Figure R6. Alfa-step depth profile of the etched GaAs photoanode film with time.

2. The etching time should be optimized according to the overall PEC performance. What is the optimal etching time of the Et-GaAs for maximizing the PEC performance (highest photocurrent and smallest onset potential)?

Response: We thank the reviewer for this comment. As suggested, the effect of the etching time on PEC performance with onset potential was studied and shown in **Figure R7**. The GaAs photoanodes are etched for 10, 20, or 30 minutes at a constant current of 0.001A and are labeled as Et-GaAs-10 min, Et-GaAs-20min, and Et-GaAs-30min, respectively. The obtained photocurrent density (J_{ph}) at 0 and 1.23 V_{RHE} and the onset potential of each photoelectrode are summarized in **Table R1**. Based on the high J_{ph} and reproducible results in PEC condition, we take the Et-GaAs-20min photoanode for further surface engineering. These results demonstrate that the surface etching of the photoanode film is a viable approach to modify the PEC behavior of GaAs-based photoelectrodes.

We also observed some interesting points based on the LSV results as shown in **Figure R7**. The etched samples show the positive shift of onset potential, compared to that of bare GaAs film, closely related to the initial surface kinetic barrier, due to the formation of native oxide during the etching process. In addition, the LSV curve has the same shape for the Et-GaAs-10min and 20

min, in which the difference of J_{ph} can be explained by the depth of surface etching sufficiently affecting the surface features such as porosity, roughness and surface area etc. Conversely, the complete opposite trend is remarkably noticed on the Et-GaAs-30min film, coming from the high density of surface native oxides and totally impeding the charge transfer kinetics at the interface.

The above discussions and the corresponding data (**Table S1 and Figure S4**) are updated in the revised manuscript.

Figure R7. LSV curves of GaAs, Et-GaAs-10 min, 20 min, and 30 min photoanode film with the magnified onset potential (V_{on}) in the right part.

Table R1. Summary of onset potential, J_{ph} at 0 and 1.23 V_{RHE} of GaAs, Et-GaAs-10 min, Et-GaAs-20min, and Et-GaAs-30min films, respectively.

	onset potential (V_{RHE})	J_{ph} at 0 V_{RHE}	J_{ph} at 1.23 V_{RHE}
GaAs	-0.364	10.21	20.3
Et-GaAs-10min	-0.323	11.1	23.61
Et-GaAs-20min	-0.321	13.5	27.38
Et-GaAs-30min	-0.060	0.11	28.29

3. The Nyquist plots and corresponding bode plots of GaAs, Et-GaAs, Et-GaAs/TiO₂, and Et-GaAs/TiO₂/NiPi photoanodes should be given together with all simulated parameters, which needs to be clearly explained.

Response: We thank the reviewer for this comment. Following the reviewer's suggestion, all Nyquist plots and corresponding Bode plots for GaAs, Et-GaAs, Et-GaAs/TiO₂, and Et-GaAs/TiO₂/NiPi photoanodes were plotted as shown in **Figure R8** and the simulated parameters are shown in **Table R2**. A revised version of these figures (**Figure 4(d)** and **Figure S12**) is included in the manuscript.

Figure R8. (a) Nyquist plot measured under open-circuit voltage under illumination, (b) bode plots and magnitude of the impedance verses logarithmic frequency.

Table R2. Fitted parameters of each sample using the suggested equivalent circuit in inset of Figure R8(a).

Photoanode	R_s (Ω)	R_{SC} (Ω)	R_{CT} (Ω)	CPE1(Q1) (nF)	CPE2(Q2) (nF)	n1	n2
GaAs	3	15.8k	22.5k	174.51	476.8	0.85	0.74
Et-GaAs	5	7.02k	-	79.60	-	0.92	-
Et-GaAs/TiO ₂	4.5	3.70k	-	68.72	-	0.94	-
Et-GaAs/TiO ₂ /Ni-Pi	4.7	1.47k	0.95k	42.2	147.5	0.88	0.65

Here we provide more discussion about the Nyquist plot. Nyquist plot with a single semicircle of Et-GaAs and Et-GaAs/TiO₂ films represents a simplified Randles circuit consisting of R_s , R_{CT} , and CPE connected in parallel with R_{CT} . R_s is associated with the resistance of the electrolyte, current collector, and other components in the system. R_{CT} represents the resistance associated with the charge transfer process at the electrode/electrolyte interface. The CPE is used instead of a pure capacitance in equivalent circuit models because it better represents the non-ideal capacitive behavior of the system. The Nyquist plot of two semicircles indicates the presence of two distinct time constants in GaAs and ET-GaAs/TiO₂ films. In this case, the modified Randles circuit can be used, which includes an additional parallel combination of resistance as well as capacitance (CPE1 and CPE2) being the constant phase elements with respective parameters of Q1, n1 and Q2, n2. The resistors (R_{SC} , R_{CT}) in the equivalent circuit represent the different charge transfer resistances associated with different processes in the photoanode. These resistances can be elucidated by the recombination of photo-generated electron-hole pairs, the charge transfer kinetics at the semiconductor/electrolyte interface, and the transport of charge carriers through the bulk of the semiconductor. In a Nyquist plot, the semicircle at the high-frequency region is closely related to the bulk charge transport in a photoelectrode film because the charge transport in the bulk of the photoelectrode film is associated with resistive and capacitive effects that occur on a fast time

scale. The low-frequency semicircle, on the other hand, is more related to the interfacial charge transfer, which is governed by the photoelectrode/electrolyte interface, including the charge transfer resistance, double-layer capacitance, and surface states. The capacitances are related to the space charge layer at the semiconductor/electrolyte interface, the interfacial capacitance, and the bulk capacitance of the semiconductor. However, it is important to note that the presence of two semicircles does not necessarily indicate the presence of two materials. Sometimes, two semicircles in a Nyquist plot can appear by the presence of a double-layer capacitance or a redox-active species. In addition, Bode plots represent the EIS data consisting of two separate plots including the magnitude of the impedance ($|Z|$) vs. frequency (logarithmic scale) and the phase angle (θ) vs. frequency (logarithmic scale). The Bode plots can help to identify the frequency range where certain photoelectrochemical processes dominate, such as charge transfer or mass transport. Magnitude of the impedance ($|Z|$) vs. frequency plot helps to identify the overall impedance and its variation with frequency. In general, a high impedance at low frequencies often indicates the diffusion or mass transport limitations, while a lower impedance at higher frequencies suggests the faster charge transfer process. The phase angle plot distinguishes between capacitive and resistive behaviors and determines the dominant processes at different frequency ranges. That is, a phase angle near 0° indicates resistive behavior (e.g., charge transfer), while a phase angle near -90° suggests capacitive behavior (e.g., double-layer charging or mass transport). As the frequency increases, the impedance magnitude decreases, and the phase angle starts to shift. When the phase angle reaches -45° , it typically indicates the onset of charge transfer processes, such as electron transfer between the photoelectrode and the electrolyte.

4. What is the optimal thickness of TiO_2 for reaching the maximum photocurrent density and the stability? Does the TiO_2 layer coated on the Et-GaAs conformally or not? It is hard to see it from the elemental mapping results.

Response: We thank the reviewer for this comment. The optimal thickness of TiO_2 layer is about $\sim 30 (\pm 5)$ nm for reaching the maximum photocurrent density and stability. **Figure R9** reveals the surface morphology of the Et-GaAs/ TiO_2 film to confirm the uniformly deposited TiO_2 nanoparticles on Et-GaAs film. Moreover, to understand the uniform coating and exact thickness over the large area of the GaAs photoanode, the α -step profilometer is used and the height profile indicates the thickness of the TiO_2 on GaAs film (**Figure R10**). To understand the reproducible results to measure the thickness of TiO_2 layer, we tested 3 batches of samples as indicated. Its thickness is estimated to be $\sim 30 (\pm 10)$ nm based on the height profile of TiO_2 on the planar GaAs photoanode. To get an exact thickness result of thin TiO_2 layer, the planar GaAs film was used.

Figure R9. (a-b) Surface FE-SEM views of Et-GaAs/TiO₂ film.

Figure R10. Alpha-step profiles of planar GaAs/TiO₂ film.

5. What is the faradaic efficiencies of the GaAs, Et-GaAs, Et-GaAs/TiO₂, and Et-GaAs/TiO₂/NiPi photoanodes? Is there a side photocurrent due to the photo-corrosion effect?

Response: We thank the reviewer for this comment. As suggested, we measured the gas evolution rate and calculated the faradaic efficiencies (FEs) for GaAs, Et-GaAs, Et-GaAs/TiO₂, and Et-GaAs/TiO₂/NiPi photoanodes as shown in **Figure R11**.

Figure R11. O₂ evolution rate corresponding faradic efficiency of the GaAs, Et-GaAs, Et-GaAs/TiO₂ and Et-GaAs/TiO₂/NiPi photoanode as a function of time.

As shown in **Figure R11**, the FE(%) of GaAs film decreased over time, subsequently corresponding to the photo-corrosion effect. In the case of photo-corrosion, the excited electrons or holes can react with the electrolyte or the semiconductor itself, leading to the formation of reactive byproducts such as hydroxyl radicals or other oxygen species or defects that can reduce the PEC performance. These side reactions (e.g., non-conductive native oxide or non-faradic charging process at the interface etc.) induce to a side current that competes with the desired photocurrent, leading to lower FE. Also, this dark current can be caused by the formation of a potential barrier at the interface between the corroded and non-corroded parts of the material. The potential barrier may act as a barrier for the electron flow, making the photocurrent pathway around the corroded area.

In the case of Et-GaAs film, the FE (%) is significant at the initial reaction time due to the high surface area effect. As time is going, the etched GaAs photoanodes are vulnerable to photo-corrosion, which can degrade their PEC performance over time. Etching can increase the surface area of the GaAs photoanode, which provides more sites for the electrolyte to react with the photoelectrode, leading to an increased susceptibility to photo-corrosion. Therefore, minimizing the photo-corrosion effect is important for improving the PEC performance and stability. However, there is no meaning degradation observed in the photoelectrodes such as Et-GaAs/TiO₂, and Et-GaAs/TiO₂/NiPi photoanodes due to the surface passivation effect, surely demonstrating the stable PEC performance and co-catalyst boosts the rapid charge transfer reaction at the surface interface. The corresponding figures (**Figure S13**) and discussion are updated in the manuscript.

6. The photovoltage was measured to be 0.95 V. However, for a self-driven water splitting in the 2-electrode configuration without external bias, a minimum photovoltage of 1.23 V+ overpotentials to ensure fast kinetic reactions (~0.4 V) is required. With such a small photovoltage

of 0.95 V, how can the Et-GaAs/TiO₂/Ni-Pi//Ni-Pi tandem cell drive the unassisted bias-free water splitting with J_{ph} of ~10.5 mA·cm⁻² and the solar-to-hydrogen conversion efficiency of 10.2%?

Response: We thank the reviewer for this comment. As suggested, we have carefully checked the tandem configuration and the corresponding results and discussion are updated in the manuscript. In our PEC cells, the optimized Et-GaAs/TiO₂/Ni-Pi//Ni-Pi photoanode gives the photovoltage ~0.95 V_{RHE} and it is not sufficient for the high efficiency because of potential loss made by the charge-transfer overpotential at electrocatalyst and ohmic loss in the anode, cathode and membrane part. In order to make the sufficient photovoltage to drive un-assisted water splitting, the dual Et-GaAs/TiO₂/Ni-Pi//Ni-Pi films are connected in series and the suggested PEC performance can be achieved, although the photocurrent is reduced overall as increasing the series connection, compared to single cell, ascribed to the charge process and mainly ohmic loss occurring in the H-type cell divided by membrane (**Figure R12(a)**). The photoanode area is 0.2 cm² in both films.

Note that the STH (%) efficiency is achieved ~8.5% up to 5h of the device performance and slightly reduced and reached 7% at ~30 h of device operation under harsh environment (**Figure R12(b)**). During the durability test, the amount of gas evolution rate is monitored and the corresponding experimental FE was calculated for H₂ (g) and O₂ (g) products (**Figure R12(c)**). And the corresponding figures (**Figure S20 and Figure 6(c, d)**) are updated in the manuscript.

Figure R12. LSV curve of Et-GaAs/TiO₂/Ni-Pi || Ni-Pi electrocatalyst in the tandem cell configuration under chopped illumination, (b) Stability of unassisted zero-bias PEC-EC tandem cell under illumination and darkness at constant time intervals in 1M NaOH and, (c) The H₂ and O₂ evolution rate with FE (%) under tandem configuration as a function of time.

Reviewer #3 (Remarks to the Author):

In the present manuscript, the authors study a GaAs/TiO₂/Ni-Pi photoanode for water oxidation under alkaline conditions. Along this study, the authors performed a detailed structural, morphological, chemical and photoelectrochemical characterization of the employed materials through a wide range of different techniques. Additionally, a novel and functional unbiased tandem cell is proposed. The manuscript is well-written and the experiments well performed.

However, even though the manuscript is of interest to the solar fuels community and is well aligned with the scope of Nature Communications, there are some points that have to be addressed before publication in order to achieve the high-quality standards required in this prestigious journal.

Response: We thank the reviewer for the overall positive evaluation of this work. We greatly appreciate the reviewer’s constructive comments to help improve the quality of our manuscript.

1. In Figure 2c, the authors show the EIS results from bare and etched-GaAs samples. The authors should clarify at which potential are these Nyquist plots acquired and clarify why they proposed this equivalent circuit. Is the GaAs sample compact? Or is it possible that some Si substrate can be exposed to the electrolyte?

Response: We thank the reviewer for this comment. All Nyquist plots for GaAs, Et-GaAs, Et-GaAs/TiO₂, and Et-GaAs/TiO₂/NiPi photoanodes were measured under the open-circuit potential as shown in **Figure R13** and the simulated parameters are shown in **Table R3**. The related information is discussed (below) and updated in the main manuscript.

Figure R13. Nyquist plot measured under open-circuit voltage under illumination.

Table R3. Fitted parameters of each sample using the suggested equivalent circuit in inset of Figure 1(a).

Photoanode	R_s (Ω)	R_{sc} (Ω)	R_{ct} (Ω)	CPE1(Q1) (nF)	CPE2(Q2) (nF)	n1	n2
GaAs	3	15.8k	22.5k	174.51	476.8	0.85	0.74
Et-GaAs	5	7.02k	-	79.60	-	0.92	-
Et-GaAs/TiO ₂	4.5	3.70k	-	68.72	-	0.94	-
Et-GaAs/TiO ₂ /Ni-Pi	4.7	1.47k	0.95k	42.2	147.5	0.88	0.65

Here we provide more discussion about the Nyquist plot. Nyquist plot with a single semicircle of Et-GaAs and Et-GaAs/TiO₂ films represents a simplified Randles circuit consisting of R_s , R_{CT} , and CPE connected in parallel with R_{CT} . R_s is associated with the resistance of the electrolyte, current collector, and other components in the system. R_{CT} represents the resistance associated with the charge transfer process at the electrode/electrolyte interface. The CPE is used instead of a pure capacitance in equivalent circuit models because it better represents the non-ideal capacitive behavior of the system. The Nyquist plot of two semicircles indicates the presence of two distinct time constants in GaAs and ET-GaAs/TiO₂ films. In this case, the modified Randles circuit can be used, which includes an additional parallel combination of resistance as well as capacitance (CPE1 and CPE2) being the constant phase elements with respective parameters of Q_1, n_1 and Q_2, n_2 . The resistors (R_{sc}, R_{CT}) in the equivalent circuit represent the different charge transfer resistances associated with different processes in the photoanode. These resistances can be elucidated by the recombination of photo-generated electron-hole pairs, the charge transfer kinetics at the semiconductor/electrolyte interface, and the transport of charge carriers through the bulk of the semiconductor. In a Nyquist plot, the semicircle at the high-frequency region is closely related to the bulk charge transport in a photoelectrode film because the charge transport in the bulk of the photoelectrode film is associated with resistive and capacitive effects that occur on a fast time scale. The low-frequency semicircle, on the other hand, is more related to the interfacial charge transfer, which is governed by the photoelectrode/electrolyte interface, including the charge transfer resistance, double-layer capacitance, and surface states. The capacitances are related to the space charge layer at the semiconductor/electrolyte interface, the interfacial capacitance, and the bulk capacitance of the semiconductor. However, it is important to note that the presence of two semicircles does not necessarily indicate the presence of two materials. Sometimes, two semicircles in a Nyquist plot can appear by the presence of a double-layer capacitance or a redox-active species.

Finally, we note that EIS was measured at OCV condition under the illumination and the GaAs sample is compact, as below cross-sectional FE-SEM view (**Figure R14**) of the bulk GaAs sample. Also, the circumstance where the GaAs substrate is exposed to the electrolyte may be impossible.

Figure R14. Cross-sectional FE-SEM image of GaAs film.

2. In Figure 4f, the authors show the band energy diagram of the complete photoanode. First of all, how were all these positions determined? On the other hand, how is possible that a 30 nm TiO₂ layer is not blocking the charge transfer from GaAs to the Ni-Pi co-catalysts with such band alignment? The authors must clarify this issue. If the role of the TiO₂ layer is passivation and protection, the thickness should be much lower, especially with such thermodynamic impediment. The authors have to clarify this point deeply. It is crucial.

Response: We thank the reviewer for this comment. There are multiple questions in this comment. We are grateful to have this opportunity to clarify these questions. We provide our responses as detailed below.

Figure R15. (a) UV-Vis absorption spectra including the Tauc plots (inset of (a)), (b) Mott-Schottky plots and (c) Energy band diagram based on GaAs, TiO₂, Ni-Pi co-catalyst and electrolyte.

- Combining UV-Vis spectroscopy and Mott-Schottky analysis data help to construct a band energy diagram for a photoanode. From the UV-Vis spectroscopy to use Tauc plot, the optical bandgap (E_g) of GaAs and TiO₂ was measured. Secondly, the flat band potential (V_{fb}) from the Mott-Schottky analysis was achieved by the reciprocal of the square of the capacitance ($1/C^2$) versus the applied potential (V_{RHE}) with fitting the data to a linear function. The V_{fb} is an approximation of the conduction band minimum (CBM) versus the reference electrode. Accordingly, we can calculate the valence band maximum (VBM) using above E_g by adding the energy difference to the CBM, shown in **Figure R15**. At last, the redox potential of GaAs was measured using $Fe(CN)_6^{3-}/Fe(CN)_6^{4-}$ redox couples containing electrolyte in the cyclic voltammetry and the cathodic/anodic peak potential of GaAs was found to be around -0.4 V and 0.4 V vs. sat. Ag/AgCl electrode, respectively, with a scan rate of 50 mV/s. Furthermore, this redox potential was used in the Mott-Schottky analysis to calculate the absolute potential scale. Following the equation (1-3);

$$E_{vb} = E_{redox} + 4.44 + V_{fb} \text{ ----- (1)}$$

$$E_{cb} = E_{vb} - E_g \text{ ----- (2)}$$

Convert the redox peak potentials to the absolute energy scale using the normal hydrogen electrode (NHE), the equation for the conversion is:

$$E_{abs} = E_{ref} + E^\circ(NHE) \text{ ----- (3)}$$

where E_{abs} is the absolute potential, E_{ref} is the redox peak potential measured with respect to the reference electrode, and $E^{\circ}(\text{NHE})$. In here, we calculated the valence band and conduction band positions, which can be used to construct the band energy diagram of the photoanode or other semiconductor materials.

- On the 30 nm TiO_2 thickness, it seems like that it is still possible for charge transfer to occur from the photoanode to the co-catalysts. Here, we provide some validated points for this phenomenon make it clear in the PEC system. The nanoscale TiO_2 particle on the GaAs can allow the photogenerated charge carrier by tunneling effect through it. This allows for charge transfer to occur even if the layer is seemingly insulating. Also, the imperfections, such as vacancies, interstitials, or grain boundaries, can create pathways for charge transfer. These defects can act as conducting pathways, facilitating the movement of charge carriers through the layer. To clarify the defect state of Ti^{3+} in the TiO_2 layer, the XPS analyses of Et-GaAs/ TiO_2 film were performed as a function of annealing time (**Figure R16(a)**), confirming the presence of Ti^{3+} state and their content is varied depending on the annealing time. In detail, in **Figure R16(a)**, the Et-GaAs/ TiO_2 -250°C, Et-GaAs/ TiO_2 -300°C and Et-GaAs/ TiO_2 -350°C films exhibited the +4 oxidation state related to $\text{Ti}2p_{3/2}$ and $\text{Ti}2p_{1/2}$, positioned at binding energies (BEs) of 459.01 eV, 458.9 eV, 458.83 eV and 464.9 eV, 464.6 eV, 464.4 eV, respectively, to prove the formation of TiO_2 phase, whereas the +3 oxidation state related to $\text{Ti}2p_{1/2}$ is also observed at BEs of 457.80 eV, 457.60 eV and 457.70 eV, respectively, signifying that a large amount of $\text{Ti}_2\text{O}_3(+3)$ is formed. These defect states in the crystalline TiO_2 are mainly responsible for the charge transfer toward the surface to drive the significant PEC water oxidation. In here, the density of Ti^{3+} peak is low at 250°C annealing and the increase of the annealing temperature to 300°C is activating more Ti^{3+} ions and increase the peak density, but decreased at 350°C, pointing out that the annealing temperature is enough high to cause Ti^{3+} ions to be reoxidized to Ti^{4+} ions, or if the thermal energy at 350°C may cause other competing reactions such as diffusion or clustering that reduce Ti^{3+} ions. Furthermore, to understand the quantitative generation of oxygen vacancies (O_v), electron paramagnetic resonance (EPR) or electron spin resonance (ESR) spectroscopy is recorded to survey the presence of oxygen vacancies or Ti^{3+} with the paramagnetic species containing unpaired electrons as shown in **Figure R16(b)**. Here, Et-GaAs/ TiO_2 /Ni-Pi photoanode films were prepared under different annealing temperature conditions, and their EPR characteristics were compared at room temperature, as shown in **Figure R16(b)**. Then, all films, including Et-GaAs/ TiO_2 -250°C, Et-GaAs/ TiO_2 -300°C and Et-GaAs/ TiO_2 -350°C films showed strong resonance signals with g factor at 2.002 and 1.983, implying that oxygen vacancy sites and electrons trapped on oxygen vacancies were closely associated with Ti^{3+} on oxygen vacancies. Considering that the intensity variation of the EPR signal represents the presence of different magnitudes of oxygen vacancies, the EPR signal of Et-GaAs/ TiO_2 (300°C) film exhibits a more intense and broader peak than the others. The results demonstrate that O_v at the ET-GaAs/ TiO_2 -300°C film may possess the optimum O_v concentration, contributing to the highest PEC performance.

- This can be concluded that the effective charge transfer is facilitated in the optimized Et-GaAs/TiO₂(300°C)/Ni-Pi photoanode through the defective conducting TiO₂ layer to the Ni-Pi surface.

Figure R16. (a) Core-level XPS spectra of Ti2*p*, (b) EPR spectra of Et-GaAs/TiO₂-250°C, Et-GaAs/TiO₂-300°C and Et-GaAs/TiO₂-350°C films and (c) Charge transfer phenomenon of photo-generated carriers in the Et-GaAs/TiO₂/Ni-Pi photoanode film.

- In addition, the incomplete pinholes at the GaAs/TiO₂ interface can be filled with Ni-Pi co-catalyst layer by electrodeposition and there might be direct contact points between the photoanode and the co-catalysts. These points would allow for efficient charge transfer without having to pass through the TiO₂ layer. Furthermore, the morphology of the TiO₂ layer can also influence the charge transfer process. In here, the compact nanoparticles can increase the contact area between the photoanode and the co-catalysts, thus facilitating charge transfer. TiO₂ is relatively have high electron mobility and can move quickly all charge carriers through the TiO₂ layer, making it easier for them to be transferred to the co-catalyst. In our fabrication method, the TiO₂ layer has a high interfacial contact with both the GaAs photoanode and the Ni-Pi co-catalysts, it can facilitate the charge transfer process. Additionally, the 30 nm thick TiO₂ layer has a high surface area and porosity, it can provide more pathways for the charges to travel, making it easier for them to reach the co-catalyst. In summary, the proper alignment of the energy bands between the photoanode, TiO₂ layer, and co-catalysts enables efficiently transfer the charges. The seemingly defected TiO₂ layer (not completely dense and crystalline TiO₂ layer) can still allow for charge transfer from the photoanode to the co-catalysts due to charge tunneling, defects, incomplete coverage, high surface area, and energy band alignment. **Figure R16(c)** represents a simple charge-transfer scheme of Et-GaAs/TiO₂/Ni-Pi photoanode film, revealing that a 30 nm TiO₂ layer can act as the cascading medium for the favorable charge transfer between the GaAs photoelectrode and co-catalyst interface.

3. Also, why do they deposit the TiO₂ layer over the intrinsic passivating layer of 10 nm that appears after the etching process?

Response: We thank the reviewer for this comment. In this work, the electrochemical etching technique of bare GaAs film induces the porous and rough surface nanotexture to reduce light reflection. This etching process causes the GaAs(OH)_x formation on the top surface region. After the annealing process, the amorphous features convert into thin native oxides such as gallium oxide (Ga₂O₃) and arsenic oxide (As₂O₃ or As₂O₅). These native oxides are unstable in the PEC condition, which can cause surface defects to provide the charge recombination site, limiting the PEC performance. In this work, it was found that such native oxides can support the passivation role of TiO₂ layer as well as contribute to the uniform TiO₂ deposition by spin-coating method, compared to the planar GaAs/TiO₂ photoanode film.

4. In figure 4c, why do the authors propose the same equivalent circuit? Do no other features appear in the Nyquist's plots? At which potential were these plots acquired?

Response: We thank the reviewer for this comment. The EIS analysis and fitted data were already performed, as shown in **Figure R13** and **Table R3**. **Figure 4c** also showed similar Nyquist plots, indicating the similar material/interface/device configuration and the same equivalent circuit was adapted to fit the raw data to get the quantitative information of each component. Herein, the EIS analysis was also carried out at OCV condition under illumination.

Reviewer #4 (Remarks to the Author):

This paper reported the use of Ni-P/TiO₂ deposited GaAs photoanodes to construct a PEC cell enabling overall solar water splitting. Stable, bias-free PEC solar water splitting with the STH efficiency of 10.2 % can be achieved. Although the topic of the work was significant with regard to the development of PEC solar water splitting, the current study failed to highlight the merits of the developed systems. The solution to the most critical issue facing PEC water oxidation was also missed. Considering the high criterion of the journal, I recommend to reject the manuscript of the present form. The following were specific comments.

Response: We greatly appreciate the reviewer's important and constructive comments. We are sorry that we didn't make several important points clear in our original manuscript as indicated by the reviewer. We have carefully considered each comment, and we believe the revised manuscript is significantly improved.

Comments:

(1) Similar approaches to passivating GaAs photoanodes for enhancing the efficiency of PEC water oxidation have been widely reported (see examples at DOI:10.1126/science.1251428; DOI: 10.1021/acscenergylett.0c02521). The authors failed to highlight the difference of the present work from those already reported, making it difficult to appreciate the merits of the present system.

Response: We thank the reviewer for this comment. We agree with the reviewer's point of view. Here, we highlight the significance of this work in comparison to similar reported GaAs photoanodes of PEC water oxidation, as detailed below:

- A simple surface etching process created porous surface features in the *n*-GaAs film, without altering the intrinsic properties of GaAs. Rather than using time-consuming, high-cost dry etching, a simple electrochemical etching method under mild conditions was adapted. This highly reproducible method can be applied to other semiconductor photoelectrodes (oxides, nitride, and sulfide).
- As a part of our study, we developed a simple spin-coating method using titanium butoxide with an optimum concentration and well-defined spin-coating parameters such as precursor volume, spin speed, and spin time. Interestingly, by changing these parameters, the uniform coverage and thickness can be achieved. One can use the spin-coating method to make a wide range of passivation layers on photoelectrodes or other catalysts with these types of tunable properties.
- Moreover, TiO₂ layer is composed of tiny nanoparticles with diameter ~15 nm rather than planar layer on the GaAs photoanode. These nanoparticles can significantly improve the surface area due to their tiny size and large number of particles in a given volume. This means that there are more particles available to interact with their surroundings, leading to a greater surface area for the same amount of material. The large surface area provided by nanoparticles increases the contact area between the electrode and the electrolyte, leading to more electrochemically active sites. In addition, the nanoparticles after post-annealing treatment can be attached strongly on the surface of the Et-GaAs, thereby improving the passivation effect. This can potentially lower the direct contact of the electrolyte to sensitive GaAs surface needed for the photocorrosion reaction, making more cost-effective and simple process.
- The defective crystalline TiO₂ protection layer on the surface of a photoanode can facilitate the transfer of the photogenerated carrier to the electrode/electrolyte interface. This is because the defects in the crystalline structure of TiO₂ can create additional energy levels within the bandgap of the material, which can act as trap sites for the photogenerated carriers. As a result, the carriers are more likely to be captured by these trap sites and subsequently transferred to the electrode/electrolyte interface, where they can participate in the water oxidation reaction. This can lead to improved efficiency in PEC processes.
- To clarify the defect state of Ti³⁺ in the TiO₂ layer, the XPS analyses of Et-GaAs/TiO₂ film were performed as a function of annealing time (**Figure R17(a)**), confirming the presence of Ti³⁺ state and their content is varied depending on the annealing time. In detail, in **Figure R17(a)**, the Et-GaAs/TiO₂-250°C, Et-GaAs/TiO₂-300°C and Et-GaAs/TiO₂-350°C films exhibited the +4 oxidation state related to Ti2*p*_{3/2} and Ti2*p*_{1/2}, positioned at binding energies(BEs) of 459.01 eV, 458.9eV 458.83 eV and 464.9 eV, 464.6 eV, 464.4 eV, respectively, to prove the formation of TiO₂ phase, whereas the +3 oxidation state related to Ti2*p*_{1/2} is also observed at BEs of 457.80 eV, 457.60 eV and 457.70 eV, respectively, signifying that a large amount of Ti₂O₃(+3) is formed. These defect states in the crystalline TiO₂ are

mainly responsible for the charge transportation toward the surface to drive the significant PEC water oxidation. In here, the Ti^{3+} peak density is low at 250°C annealing and the increase of the annealing temperature to 300°C is activating more Ti^{3+} ions and increase the peak density, but decreased at 350°C, pointing out that the annealing temperature is enough high to cause Ti^{3+} ions to be reoxidized to Ti^{4+} ions, or if the thermal energy at 350°C may cause other competing reactions such as diffusion or clustering that reduce Ti^{3+} ions. Furthermore, to understand the quantitative generation of oxygen vacancies (O_v), electron paramagnetic resonance (EPR) or electron spin resonance (ESR) spectroscopy is recorded to survey the presence of oxygen vacancies or Ti^{3+} with the paramagnetic species containing unpaired electrons as shown in **Figure R17(b)**. Here, Et-GaAs/ TiO_2 /Ni-Pi photoanode films were prepared under different annealing temperature conditions, and their EPR characteristics were compared at room temperature, as shown in **Figure R17(b)**. Then, all films, including Et-GaAs/ TiO_2 -250°C, Et-GaAs/ TiO_2 -300°C and Et-GaAs/ TiO_2 -350°C films showed strong resonance signals with g factor at 2.002 and 1.983, implying that oxygen vacancy sites and electrons trapped on oxygen vacancies were closely associated with Ti^{3+} on oxygen vacancies. Considering that the intensity variation of the EPR signal represents the presence of different magnitudes of oxygen vacancies, the EPR signal of Et-GaAs/ TiO_2 (300°C) film exhibits a more intense and broader peak than the others. The results demonstrate that O_v at the ET-GaAs/ TiO_2 -300°C film may possess the optimum O_v concentration, contributing to the highest PEC performance.

Figure R17. (a) Core-level XPS spectra of $Ti2p$, (b) EPR spectra of Et-GaAs/ TiO_2 -250°C, Et-GaAs/ TiO_2 -300°C and Et-GaAs/ TiO_2 -350°C films and (c) Charge transfer phenomenon of photo-generated carriers in the Et-GaAs/ TiO_2 /Ni-Pi photoanode film.

(2) Similar to the electrolytic OER, PEC water oxidation in acid electrolyte also imposes a much great challenge to limit the large-scale implementation of the economically viable photoanodes. The authors should also examine the practice of the current photoanode in acid electrolyte.

Response: Thank you for the suggestion to improve the manuscript. **Figure R18** shows LSV curve of GaAs, Et-GaAs, Et-GaAs/TiO₂ and Et-GaAs/TiO₂/Ni-Pi photoelectrodes under the strong acidic solution. Considering that the proton (H⁺) mobility is higher in acidic media, which facilitates faster charge transfer and reaction rates, the fast PEC OER can lower the overpotentials, leading to higher efficiency and lower energy consumption. The obtained photocurrent density (J_{ph}) at 0, 1.23 V_{RHE} and the onset potential of the photoelectrodes in acidic condition in **Table R4**.

Figure R18. LSV response of the photoanode films at the 0.5M H₂SO₄ electrolyte.

Table R4. Summary of onset potential (V_{RHE}), J_{ph} at 0 and 1.23 V_{RHE} of GaAs, Et-GaAs, Et-GaAs/TiO₂ and Et-GaAs/TiO₂/Ni-Pi photoelectrodes.

	onset potential (V _{RHE})	J _{ph} at 0 V _{RHE}	J _{ph} at 1.23 V _{RHE}
GaAs	-0.24	0.9	22
Et-GaAs	-0.18	1.45	31
Et-GaAs/TiO ₂	-0.15	7.9	27.4
Et-GaAs/TiO ₂ /Ni-Pi	-0.11	14.46	36.3

(3) In Fig 1D, the increase in absorbance for Et-GaAs across 400 to 900 nm region was accompanied by the increase in baseline intensity. This outcome suggested that significant light scattering occurred for Et-GaAs, which should not be confounded with the increase in photo-absorption.

Response: We thank the reviewer for this comment. Etching can create a textured surface with micro- or nano-scale features. These features can scatter the light, causing it to travel longer paths within the semiconductor. This increased path length enhances the probability of photon absorption, enabling to improve the overall light absorption. Also, surface etching increases the effective surface area of the GaAs, providing more opportunities for light absorption. Considering

the nano-textured surface states, the diffusive reflectance spectra is better to compare the quantitative light absorption, shown in **Figure R19**. Herein, the Et-GaAs-20min film exhibited the sharply increased absorption spectra around 900 nm and there is no remarkably observed baseline change because it comes from the reflection or back-scattering of light by a material. Therefore, in the case of Et-GaAs, the increase in baseline intensity across the 400 to 900 nm region in the absorption spectra could indeed come from the significant light scattering.

Figure R19. Diffusive reflectance spectra of the GaAs and Et-GaAs photoanode films.

(4) A table summarizing the current advancement of the-state-of-the-art GaAs-based PEC systems and tandem cells ever reported should be provided to enable a global performance comparison.

Response: We thank the reviewer for this comment. As suggested, we summarized the recently updated GaAs based PEC performance. The summary is based on the fabrication method, photocurrent density (J_{ph}), and stability, as shown in **Table R5**.

Table R5. Summary of the recently updated GaAs based PEC performance in terms of fabrication method, photocurrent density (J_{ph}), electrolyte, and stability.

GaAs based PEC cell					
Photoanode	Fabrication	J_{ph} ($\text{mA}\cdot\text{cm}^{-2}$)	Electrolyte	Stability	Ref.
GaAs/TiO ₂ /Ni core-shell nanorods	GLAD	12.87 $\text{mA}\cdot\text{cm}^{-2}$	1M NaOH	3 h	1
GaAs/Ga ₂ S ₃ heterostructure	Sulfurization	12.87 $\text{mA}\cdot\text{cm}^{-2}$	0.1 M Na ₂ SO ₄	500 s	2
GaAs(100)/single layer of graphene	Vertical gradient-freeze method, CVD	25.8 $\text{mA}\cdot\text{cm}^{-2}$	CH ₃ CN-Fc ⁺⁰	8 h	3
GaAs/Polythiophene	Electrodeposition	3.52 $\text{mA}\cdot\text{cm}^{-2}$	0.1M K ₃ Fe(CN) ₆ , 0.25M K ₄ Fe(CN) ₆	100 h	4
GaAs NW arrays/NiO _x	MOCVD, E-beam Lithography, ALD	11.1 $\text{mA}\cdot\text{cm}^{-2}$	Fc/Fc ⁺ redox couple	-	5
GaAs/TiO ₂ /Ni	ALD	14.3 $\text{mA}\cdot\text{cm}^{-2}$	1M KOH	100 h	6

GaAs/Ni-B	MOCVD, Photo-assisted Electrodeposition	20 mA·cm ⁻²	0.1M KOH	22 h	7
GaAs/Ni	Electrodeposition	9.2 mA·cm ⁻²	K ₃ Fe(CN) ₆ /K ₄ Fe(CN) ₆	300s	8
GaAs/a-TiO ₂ /NiO _x	MOCVD, ALD, Sputtering	8.3 mA·cm ⁻²	1M KOH	600 h	9
TiN/N-TiO ₂ /ITO/GaAs	RF sputtering, NH ₃ Plasma, PECVD	17.82 mA·cm ⁻²	1M KOH	6 h	10
GaAs/Ir	Spin-coating	18 mA·cm ⁻²	1M H ₂ SO ₄	-	11
Et-GaAs/TiO ₂ /Ni-Pi	Electrochemical etching, Spin-coating and Electrodeposition	25 mA·cm ⁻²	1M NaOH	~110 h (1-Sun) ~200 h (0.5 Sun)	This work
GaAs-based Tandem cell					
Tandem cell	Fabrication	Configuration	STH (%)	stability	Ref.
GaAs/InGaAsP WO ₃ /BiVO ₄	Electrochemical deposition	PV-PEC	8.1%	1h	12
InGaP GaAs	MOCVD, Electrodeposition	PEC	~9%	150 h	13
In _{0.25} Ga _{0.75} N Pt	plasma-assisted molecular beam epitaxy	PEC-EC	3.4%	~300 h	14
BiVO ₄ p ⁺ n-GaAs _{1-x} Pt _x	Spray pyrolysis	PV-PEC	1.8%	3h	15
Et-GaAs/TiO ₂ /Ni-Pi Ni-Pi@Ni foam	Electrochemical etching, Spin-coating and Electrodeposition	PEC-EC	9.5%	~35h	This work

Reference:

- 1) Alqahtani, M., Kafizas, A., Sathasivam, S., Ebaid, M., Cui, F., Alyamani, A., & Wu, J. (2020). A Hierarchical 3D TiO₂/Ni Nanostructure as an Efficient Hole-Extraction and Protection Layer for GaAs Photoanodes. *ChemSusChem*, 13(22), 6028-6036.
- 2) Liu, H. F., Antwi, K. A., Chua, C. S., Huang, J., Chua, S. J., & Chi, D. Z. (2014). Epitaxial synthesis, band offset, and photoelectrochemical properties of cubic Ga₂S₃ thin films on GaAs (111) substrates. *ECS Solid State Letters*, 3(11), P131.
- 3) Yang, F., Nielander, A. C., Grimm, R. L., & Lewis, N. S. (2016). Photoelectrochemical behavior of n-type GaAs (100) electrodes coated by a single layer of graphene. *The Journal of Physical Chemistry C*, 120(13), 6989-6995.
- 4) Horowitz, G., & Garnier, F. (1985). Long-Term Stabilization of Polythiophene-Protected n-GaAs Photoanodes in Aqueous Solution. *Journal of the Electrochemical Society*, 132(3), 634.
- 5) Zeng, J., Xu, X., Parameshwaran, V., Baker, J., Bent, S., Wong, H. S. P., & Clemens, B. (2018). Photoelectrochemical water oxidation by GaAs nanowire arrays protected with atomic layer deposited NiO_x electrocatalysts. *Journal of Electronic Materials*, 47, 932-937.
- 6) Hu, S., Shaner, M. R., Beardslee, J. A., Lichterman, M., Brunshwig, B. S., & Lewis, N. S. (2014). Amorphous TiO₂ coatings stabilize Si, GaAs, and GaP photoanodes for efficient water oxidation. *Science*, 344(6187), 1005-1009.
- 7) Jiang, C., Wu, J., Moniz, S. J., Guo, D., Tang, M., Jiang, Q., Tang, J. (2019). Stabilization of GaAs photoanodes by in situ deposition of nickel-borate surface catalysts as hole trapping sites. *Sustainable Energy & Fuels*, 3(3), 814-822.

- 8) Xu, Y., Ahmed, R., Zheng, J., Hoglund, E. R., Lin, Q., Berretti, E., Zangari, G. (2020). Photoelectrochemistry of Self-Limiting Electrodeposition of Ni Film onto GaAs. *Small*, 16(39), 2003112.
- 9) Shen, X., Yao, M., Sun, K., Zhao, T., He, Y., Chi, C. Y., Hu, S. (2020). Defect-tolerant TiO₂-coated and discretized photoanodes for > 600 h of stable photoelectrochemical water oxidation. *ACS Energy Letters*, 6(1), 193-200.
- 10) Choi, K., Bang, J., kyu Moon, I., Kim, K., Oh, J. (2020). Enhanced photoelectrochemical efficiency and stability using nitrogen-doped TiO₂ on a GaAs photoanode. *Journal of Alloys and Compounds*, 843, 155973.
- 11) Pishgar, S., Mulvehill, M. C., Gulati, S., Sumanasekera, G. U., Spurgeon, J. M. (2021). Investigation of n-GaAs Photoanode Corrosion in Acidic Media with Various Thin Ir Cocatalyst Layers. *ACS Applied Energy Materials*, 4(10), 10799-10809.
- 12) Kosar, S., Pihosh, Y., Turkevych, I., Mawatari, K., Uemura, J., Kazoe, Y., Kitamori, T. (2016). Tandem photovoltaic-photoelectrochemical GaAs/InGaAsP-WO₃/BiVO₄ device for solar hydrogen generation. *Japanese journal of applied physics*, 55(4S), 04ES01.
- 13) Varadhan, P., Fu, H. C., Kao, Y. C., Horng, R. H., & He, J. H. (2019). An efficient and stable photoelectrochemical system with 9% solar-to-hydrogen conversion efficiency via InGaP/GaAs double junction. *Nature Communications*, 10(1), 5282.
- 14) Wang, Y., Wu, Y., Schwartz, J., Sung, S. H., Hovden, R., & Mi, Z. (2019). A single-junction cathodic approach for stable unassisted solar water splitting. *Joule*, 3(10), 2444-2456.
- 15) Lawrence, D. J., Smith, B. L., Collard, C. D., Elliott, K. A., Fakhoury, K. L., Mangold, J. D., Soyka, A. N. (2021). Monolithically-integrated BiVO₄/p⁺-n GaAs_{1-x}P_x tandem photoanodes capable of unassisted solar water splitting. *International Journal of Hydrogen Energy*, 46(2), 1642-1655.

REVIEWERS' COMMENTS

Reviewer #3 (Remarks to the Author):

The authors have successfully addressed all the comments and questions raised by the reviewers, leading to a highly improved study. Hence, I recommend the publication of the present manuscript in its current form.

Reviewer #4 (Remarks to the Author):

As stated in the original reviewer's comments, the main concern rooted in the failure of addressing the difference of the present work from those already reported in Science (DOI:10.1126/science.1251428) and ACS Energy Lett (DOI: 10.1021/acseenergylett.0c02521). The breakthrough made from the fundamental perspective is particularly important because it can enable the widespread deployment of the proposed concept/technique to other systems. Only with this justification could the research work meet the high criterion of Nature Communications. In the revised manuscript, the authors highlighted several aspects to explain the causes for the activity improvement over other similar systems. These highlights were mostly associated with the modification of fabrication processes. How to extend such an on-hand experience into a global scope is a question mark. I intend to stick to my original comment, but I would leave the final decision to the Editor.